# A bedform phase diagram for dense granular currents

Gregory Smith [1]✉, Peter Rowley[1,2,3], Rebecca Williams [1], Guido Giordano[4], Matteo Trolese [4], Aurora Silleni [4], Daniel R. Parsons [5] & Samuel Capon[2]

Pyroclastic density currents (PDCs) are a life-threatening volcanic hazard. Our understanding and hazard assessments of these flows rely on interpretations of their deposits. The occurrence of stratified layers, cross-stratification, and bedforms in these deposits has been assumed as indicative of dilute, turbulent, supercritical flows causing traction-dominated deposition. Here we show, through analogue experiments, that a variety of bedforms can be produced by denser, aerated, granular currents, including backset bedforms that are formed in waning flows by an upstream-propagating granular bore. We are able to, for the first time, define phase fields for the formation of bedforms in PDC deposits. We examine how our findings impact the understanding of bedform features in outcrop, using the example of the Pozzolane Rosse ignimbrite of the Colli Albani volcano, Italy, and thus highlight that interpretations of the formative mechanisms of these features observed in the field must be reconsidered.

[1] Department of Geography, Geology and Environment, University of Hull, Hull, UK. [2] School of Earth and Environmental Sciences, University of Portsmouth, Portsmouth, UK. [3] Department of Geography and Environmental Sciences, University of the West of England, Bristol, UK. [4] Dipartimento di Scienze, Università Roma Tre, Roma, Italia. [5] Energy and Environment Institute, University of Hull, Hull, UK. ✉email: Gregory.Smith-2016@hull.ac.uk

Particulate density currents are the largest mass transporters of sediment on the Earth's surface. Deep-sea turbidity currents deposit the largest sediment accumulations on the Earth[1], density currents emplace ejecta blankets around bolide impact craters[2] and pyroclastic density currents (PDCs) can transport thousands of cubic kilometres of volcanic material during a single event[3]. These flows also pose a major geohazard, with deep-sea turbidity currents threatening seafloor infrastructure and PDCs being responsible for over 90,000 deaths since 1600 CE[4,5]. Understanding the behaviour of these particle-laden, fast-moving currents is fundamental to decreasing the risks they pose to society.

The dynamics and depositional processes of PDCs are difficult to analyse due to their destructiveness, and the concealment of the internal dynamics by an accompanying ash cloud. Understanding of PDC behaviour, therefore, is primarily based on interpretation of the geological record preserved in sedimentary deposits[6–10], complemented by analogue and numerical modelling[11–14].

The presence and morphology of sedimentary structures, such as bedforms, in a deposit can be interpreted to tell us about the internal behaviour of the density current that formed them[15–19]. Various types of cross-stratified bedforms occur in PDC strata and are assumed to be formed by dilute, high-velocity (surge) PDCs[8,18,20–24], where tractional processes dominate in the flow-boundary zone due to the predominance of fluid turbulence as a particle support mechanism[9,11,25,26]. Denser, granular fluid-based PDCs are usually thought to be responsible for the creation of massive deposits, lacking in sedimentary structures[6,9,27,28].

Bedform-related sedimentary structures in PDC deposits include backset features (i.e., upstream-dipping beds) formed by stoss-side aggradation, similar to chute-and-pool structures and antidunes found in fluvial systems (Fig. 1a, f and Fig. 1b, d), which are generally thought to be formed under supercritical flow conditions[16,19,29,30]. Early work on such structures in PDC deposits interpreted them similarly as the result of supercritical flows[31–34]. These backset bedforms have commonly been referred to as regressive, for example by Allen[18], who interpreted them as sandwaves deposited by wet and cool pyroclastic surges. Since then regressive has been commonly used to describe stoss-aggrading features in PDC deposits, although linking this to flow conditions, rather than temperature and moisture content[21,35–37]. However, there have been attempts to introduce new terminology which does not hold the genetic connotations of antidune, chute-and-pool or sandwave. For example, Brown and Branney[38] use the term regressive bedform for a giant set of sigmoidal, upstream-dipping lenses. Douillet et al.[22] introduce the term regressive climbing dunes for bedforms which show upstream crest migration (Fig. 1c). Brand et al.[39] adopt similar terminology, using regressive dune bedforms (Fig. 1e). In this paper, we avoid using such terms, in the interests of being purely descriptive, opting instead to use backset bedforms to refer to stoss-aggrading features, which have both asymmetrical (much steeper stoss sides; Fig. 1g) or roughly symmetrical lee and stoss slopes (Fig. 1h).

Analogue modelling of dense PDCs has advanced considerably over recent years, including work focusing on the influence of pore pressure[13,40–45]. High gas pore pressure created by various mechanisms within PDCs[6,9,46–48] has been shown to be responsible for their unusually high mobility[49–51], but only recently has physical modelling reflected the sustained and variable nature of such pore pressures with distance from source[44,52].

Here, we examine the conditions which promote the growth of bedforms in aerated dense granular flows, as analogues for PDCs and their deposits. This work describes laboratory experiments, in which we use partially fluidised (aerated) fine-grained particles in a 3-m long flume (see 'Methods'). These experiments are able to simulate many behaviours of PDCs[13,43,44,52]. As the deposit aggrades from the quasi-steady currents, the growth of bedforms is recorded using a high-speed camera. We study how backset bedform features form within the dense granular currents. Deposition is triggered in the experiments as the sustained aerated flow passes into a section of the flume with a reduced or absent basal gas flux, resulting in rapid deaeration and a consequent increase in frictional forces between particles. This is not intended to represent a specific natural process but rather simulate the rapid deaeration hypothesised to occur in natural PDCs as a result of various processes, such as loss of fines, temperature drops, thinning and/or the entrainment of coarser

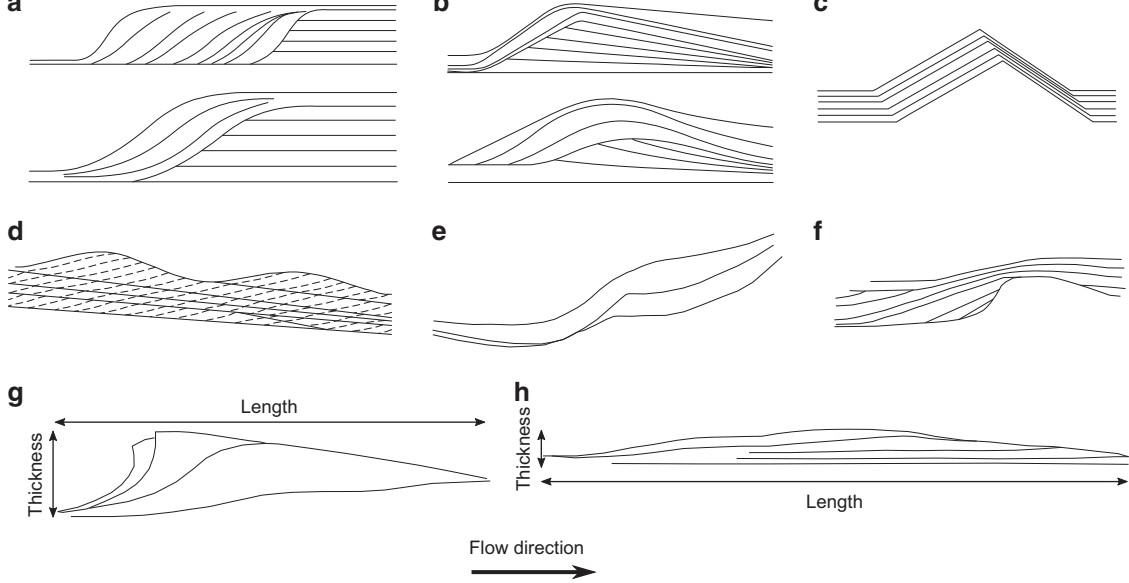

**Fig. 1 Sketches of backset bedforms in PDC and fluvial deposits. a** Chute-and-pool structures in dilute PDC deposits at Laacher See[20]. **b** Antidunes in dilute PDC deposits at Laacher See[20]. **c** Regressive dune bedform[22]. **d** Stable antidunes[30]. **e** Regressive bedform from the Proximal Bedded Deposits at Mt St Helens[39]. **f** Fluvial chute-and-pool structure[75]. **g** Steep backset bedform as described in this paper, showing length and thickness definitions. **h** Shallow backset bedform, as described in this paper.

material[45,48,53]. The initial deaeration would be accelerated by the slowing current (decreasing shear rates), and increasing inter-particle frictional forces. We are able to, for the first time, define phase fields for the formation of types of bedforms in PDC deposits using current velocity, current thickness, Froude number and Friction number. We examine how our interpretations impact on the understanding of similar features in outcrop, using the example of the Pozzolane Rosse ignimbrite of the Colli Albani volcano, Italy.

## Results

**Bedform morphology.** A range of bedforms were observed growing under a variety of flow conditions within the suite of experimental runs (see 'Methods'). We categorise these bedforms into three types (Fig. 2): planar/very shallow backset (<2°) bed-sets, backset bedforms with shallow stoss sides less than the dynamic angle of repose (<$\theta_{Dyn}$), and backset bedforms with steep (>$\theta_{Dyn}$) stoss sides. Planar bedsets, shallow backset bed-forms and steep backset bedforms are present in each deposit except one (Fig. 2e), which does not show steep backset bedforms. Both steep and shallow backset bedforms comprise a bedset of multiple (3–4) stoss-side lamina dipping at varying angles, con-verging into a single corresponding lee-side lamina (Table 1). No progressive (prograding) bedforms were observed in any of the experimental runs because our experiments are run with waning, not waxing currents.

**Bedform deposition.** The experiments began when the particles were released into the flume via trapdoor and impinged on the basal porous mesh, forming an aerated current. The leading edges of the currents were travelling at ~2 ms$^{-1}$ as they passed into the lesser/unaerated second chamber of the flume (Fig. 3a; see Sup-plementary Movie 1). The sustained currents rapidly deaerate as they pass over the second chamber of the flume, promoting deposition. Small spontaneously generated variations in the cur-rent mass flux result in minor unsteadiness in the flow over timescales in the order of 0.05 s and flow thickness variations in the order of +/−10%, hence their quasi or nearly steady nature[44]. The currents initially deposit planar or very shallow backset bedsets after the break in aeration (Fig. 3b), at velocities of ~1–1.5 ms$^{-1}$. Within 0.4–0.8 s of deposition beginning, stoss-side aggrading shallow backset bedforms are deposited above and upstream of the planar bedsets as the current velocities decrease (Fig. 3c–d). Within 1.1–1.6 s of deposition beginning, with current velocities below ~0.5 ms$^{-1}$, the upstream edge of the deposit steepens and collapses, with very steep backset bedsets deposited just prior to this, forming the stoss sides of steep backset bedforms (Fig. 3e–f). Current velocity and thickness data during deposition of the bedforms may be found in Supple-mentary Table 1.

**Velocity and thickness control on bedform formation.** Planar, shallow and steep features fall into well-defined fields on a current velocity vs. current thickness plot, suggesting that current velocity and thickness controls the sedimentary structures in the deposit (Fig. 4a). For a given current thickness, planar bedsets are deposited at higher velocities (above 0.8 ms$^{-1}$ in these experi-ments). Shallow backset bedforms are deposited at lower velo-cities, and steep backset bedforms are deposited at the lowest velocities (between 0.3 and 0.6 ms$^{-1}$ in these experiments). With increasing current thickness, higher current velocities are required to remain in the shallow bedform and planar bedform stability fields. As a result of thickening within a steady current, bedform-induced deposits of different character can be formed without a requirement for a change in flow velocity. It is

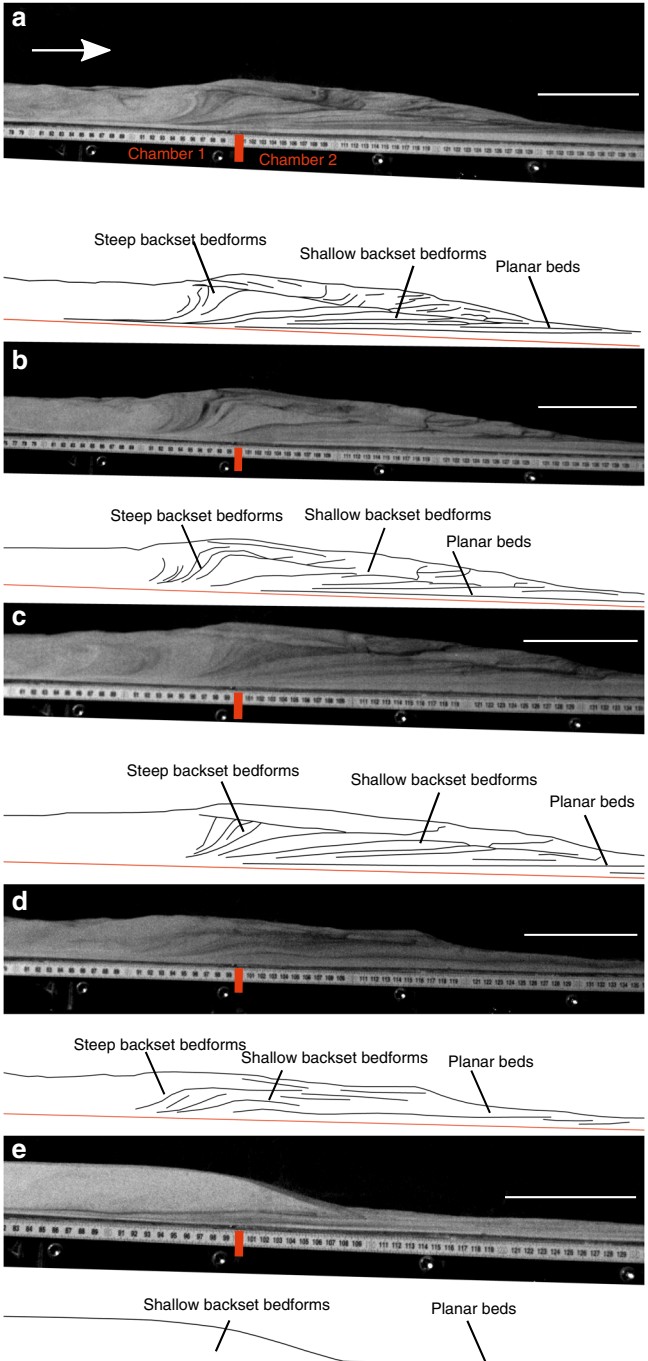

**Fig. 2 Deposits from five separate experimental runs.** Scale bar = 10 mm. **a–c** show backset bedforms deposited by currents passing above a chamber aerated at 0.93 $U_{mf}$ to one unaerated. **d** shows backset bedforms deposited by a current passing above a chamber aerated at 0.93 $U_{mf}$ to one aerated at 0.66 $U_{mf}$. **e** shows backset bedforms deposited by a current passing above a chamber aerated at 0.66 $U_{mf}$ to one aerated at 0.53 $U_{mf}$.

important to note that the deposit formed over the smallest aeration drop (0.66 $U_{mf}$ to 0.53 $U_{mf}$) does not show steep backset bedforms, and only poorly developed shallow backset bedforms, suggesting the magnitude of the aeration drop and consequent velocity changes may also have some control.

**Phase fields.** We define phase fields for the three types of bed-forms using the Froude number (*Fr*) and the Friction Number

**Table 1 Dimensions and angles of our experimental backset bedforms.**

| Bedform | Lengths (m) | Thickness (m) | Stoss angles (°) | Lee angles (°) |
| --- | --- | --- | --- | --- |
| Steep backset (Fig. 1g) | 0.18–0.4 | 0.35–0.4 | 20—overturned | <10 |
| Shallow backset (Fig. 1h) | 0.18–0.21 | 0.003–0.01 | <10 | <10 |

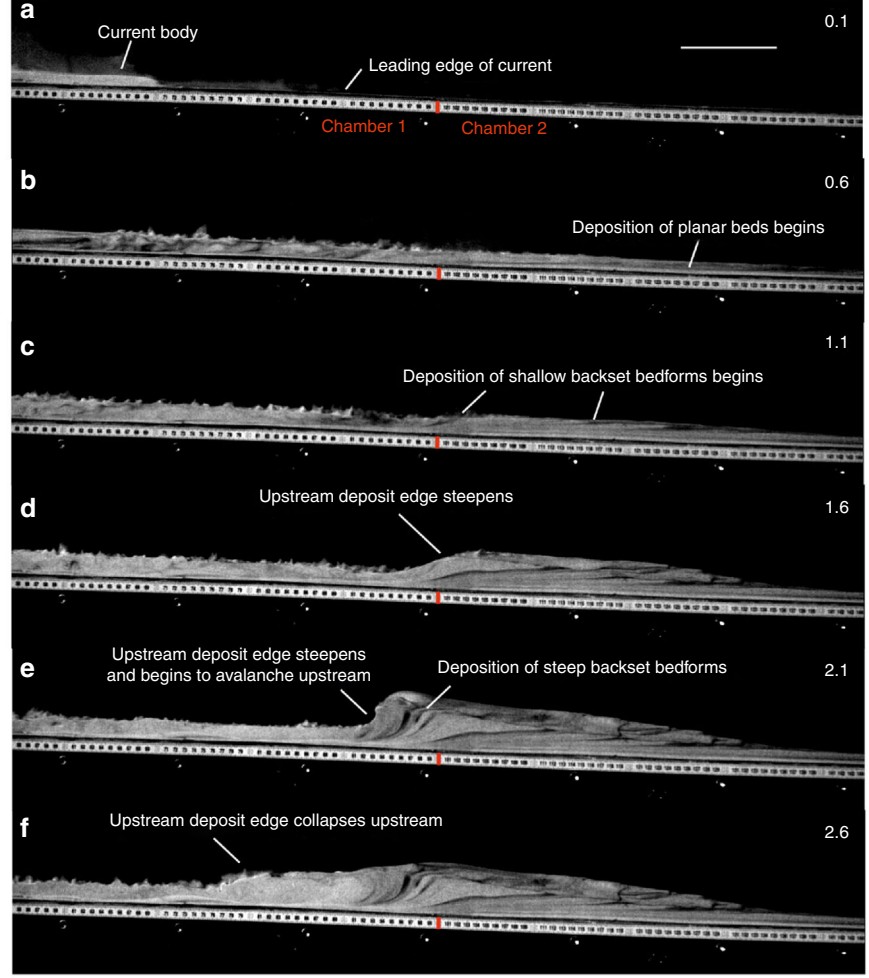

**Fig. 3 Timelapse of an experimental granular current.** Scale bar = 10 mm. **a** an experimental granular current before deposition begins. **b–c** deposition is triggered by the current passing above a chamber aerated at 0.93 U$_{mf}$ to one unaerated. **d–e** the upstream edge of the deposit steepens, forming steep backset bedforms. **f** the upstream edge of the deposit collapses. Number in the top right of the frames is the time in seconds since the current entered the first frame.

($N_F$). The Froude number ($Fr$) represents the ratio of kinetic to potential energy (Eq. (1)).

$$Fr = U/(gH)^{1/2} \qquad (1)$$

Where $U$ = current velocity, $g$ = gravity and $H$ = current thickness. The Friction number ($N_F$) is the ratio of frictional to viscous stresses and is defined as Bagnold number/Savage number[54,55]. The Savage number ($N_S$, Eq. (2)) is the ratio of collisional stress to frictional stress[55,56], and the Bagnold number ($N_B$, Eq. (3)) is the ratio of collisional stress to viscous fluid stress[55,57].

$$N_S = \frac{\left(\frac{U}{H}\right)^2 \delta^2 \rho_s}{\left(\rho_s - \rho_f\right) gH \tan\theta} \qquad (2)$$

$$N_B = \frac{\left(\frac{U}{H}\right)\delta^2 \rho_s \varphi}{(1-\varphi)\mu} \qquad (3)$$

where $\rho_s$ = particle density, $\rho_f$ = fluid density, $\delta$ = particle diameter, $\theta$ = internal friction angle, $\varphi$ = solid volume fraction and $\mu$ = fluid viscosity.

$N_S$ in these experiments range from 0.00003 to 0.03, and $N_B$ from 15 to 269. In natural PDCs, $N_S$ has been estimated to range from $10^{-8}$ to $10^{-9}$ [13], which similar to our experiments is in the frictional regime[56] despite the difference of several orders of magnitude. Our $N_B$ values overlap with those estimated for natural PDCs ($10^0$–$10^2$)[13].

Froude numbers were calculated for each tracked sediment package during its deposition. Different types of bedforms are formed under different ranges of $Fr$, with greater overlap between

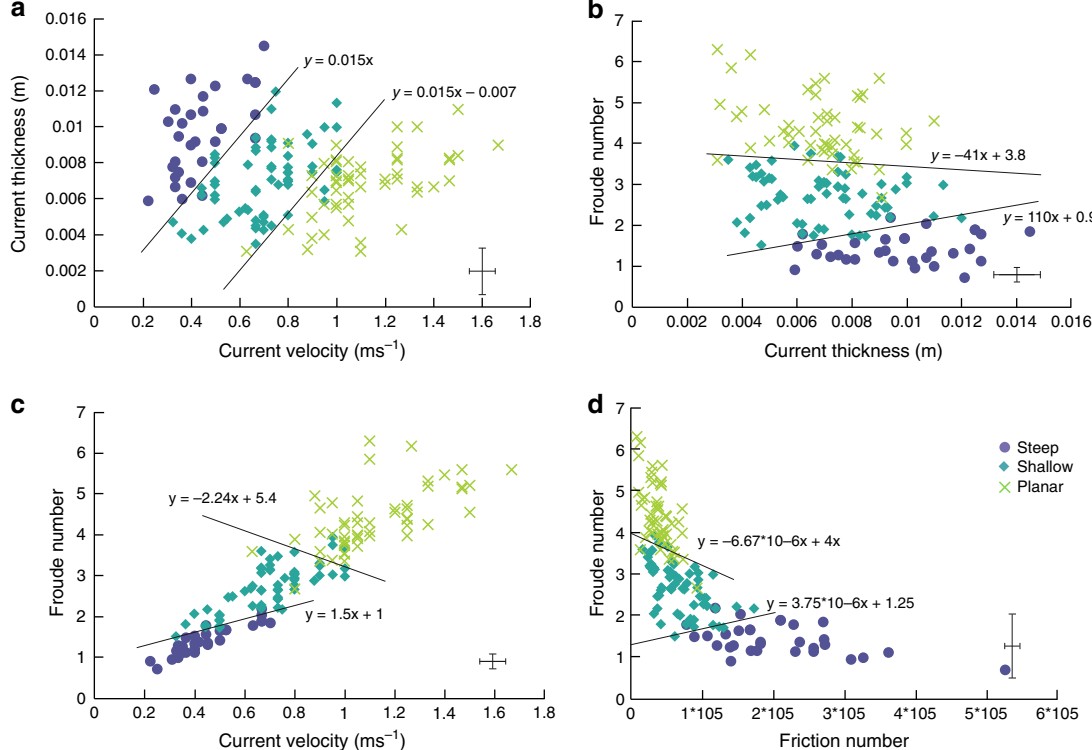

**Fig. 4 Phase diagrams showing the current conditions which control backset bedform formation, with plausible phase boundaries. a** Velocity vs. thickness. **b** Thickness vs. Froude number. **c** Velocity vs. Froude number. **d** Friction number vs. Froude number. Representative ($n = 20$) error bars are located in the bottom right of each image (±2 s.d.).

the planar bedset and shallow backset bedform fields than between the shallow and steep backset bedform fields (Fig. 4b–c). As anticipated, there is a good correlation ($R = 0.843$) between $Fr$ and velocity (Fig. 4c), but with a noticeably greater data spread at higher (>0.8 ms$^{-1}$) velocities, whereas $H$ exerts much less of a control on $Fr$ (Fig. 4b).

Planar bedsets are mostly deposited at high $Fr$ and low $N_F$, shallow backset bedforms at moderate $Fr$ and $N_F$ and steep backset bedforms at low $Fr$ and high $N_F$ (Fig. 4d). The planar-shallow-steep sequence of bedform formation can therefore be seen as recording the transition of a fast, supercritical current dominated by viscous stresses to a slower current increasingly dominated by frictional stresses.

**Similar bedforms in the field**. The Pozzolane Rosse (PR) ignimbrite covers an area of more than 1600 km$^2$ around the Colli Albani volcano, Italy[58], and has been dated ($^{40}$Ar/$^{39}$Ar) at 456 ± 3 ka[59]. It surmounts topography of 250 m to reach altitudes of 440 m[60]. The ignimbrite is generally massive, matrix-supported and poorly sorted, with a noticeable paucity in fine ash. Emplacement temperatures have been estimated to be between 630 °C and 710 °C[61].

Six samples were taken for this study from three localities (within 18–24 km of the vent; Fig. 5a) and two facies (massive, and undulated bedding as described by Giordano and Dor-onzo[62]). Grains are dominantly poorly vesicular scoria with compositions plotting in the tephrite/basanite field[63]. The grain-size distribution of all samples is dominated by lapilli-sized grains and poor in the <63-μm fraction (Fig. 5b; Supplementary Table 2), which is consistent with samples from other studies (Fig. 5c), plotting in the fines-depleted flow field of Walker[25]. Therefore, we

consider the parent PDC of the PR ignimbrite to be a good natural example of an analogue dense, granular current.

Rotating drum tests on the six samples taken from the PR (excluding grains >0.0056 m) gave static minimum ($\theta_{Smin}$), maximum ($\theta_{Smax}$) and dynamic ($\theta_{Dyn}$) angles of repose of 35.3°, 51.7° and 45.2°, respectively (Supplementary Fig. 1). Although these values are considerably higher than those obtained for the particles used in the experiments (Supplementary Fig. 2) (likely due to the variable grain size and angularity of the ignimbrite grains), the scaling remains reasonable due to the larger particle sizes in the natural materials (see Eq. (2)).

Backset bedforms are found in the undulated bedding facies in the NE sector of the PR ignimbrite, where the depositing current left the radial plain and ran up into the Apennine mountains[62]. The undulated facies transitions laterally into the massive facies of the PR on scales of hundreds of metres, and both facies have the same grain size and compositional characteristics (Fig. 5b–c), thus we interpret them to be from the same parent PDC. The bedforms in the PR share similarities with our experimental deposits (c.f. Figs. 6a and 2a–c, Figs. 6c and 2d); and measured stoss angles for both natural and experimental bedforms span the same range (Fig. 6b). The stoss layers seen in the PR backset bedforms are never overturned upstream like some of the experimental deposits. Preservation of overturned beds in natural deposits may be difficult—upstream avalanching of material from this unstable bedform may be reincorporated into a sustained current, or they may be cryptic and not easily visible in natural material. Shallow stoss-sided bedforms are found in this facies (Fig. 6d), although they tend to have greater lee (due to the greater repose angles of the material) and stoss angles than experimental examples, where both are <10° (Fig. 6b).

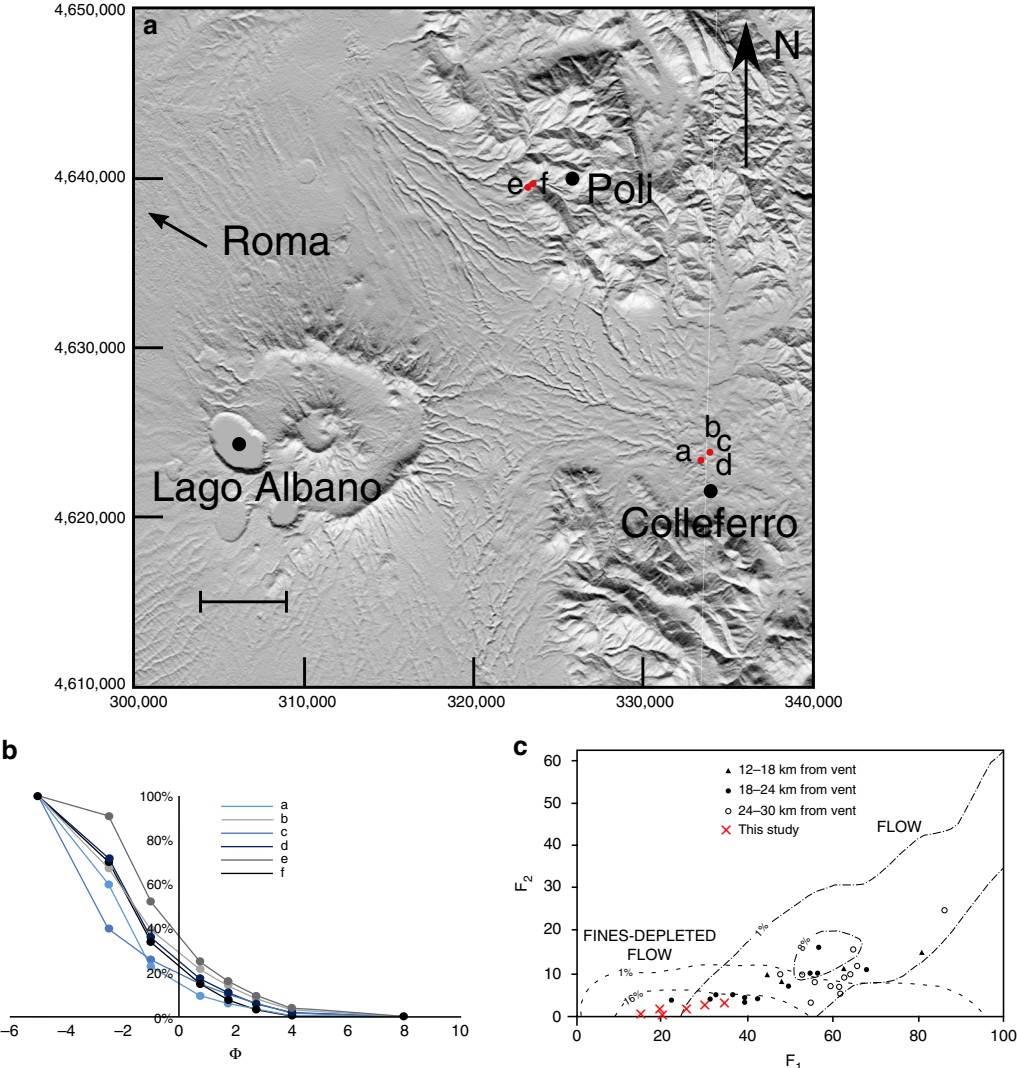

**Fig. 5 Grain-size data for samples from the Pozzolane Rosse ignimbrite. a** Map of sample locations. Scale bar = 5 km. Sample a is from the massive facies, sample b, c and d from the undulated bedding facies, and sample e and f from backset bedforms within this facies. **b** Grain-size distribution curves for samples from this study. Note the dominance of coarse grains and paucity in the <63 μm (4 φ) fraction. The grain-size data are given in Supplementary Table 2. **c** Plot of weight percentage finer than 63 μm ($F_2$) vs. weight percentage finer than 1 mm ($F_1$), after Walker[25]. Black symbols are PR ignimbrite samples from Giordano and Dobran[58], red crosses show the PR ignimbrite samples from this study.

## Discussion

Our experimental deposits consist of planar bedsets and shallow and steep backset bedforms. The existing widespread interpretation of backset features in PDC deposits is that they are a product of upper flow regime/Froude supercritical flow within dilute PDCs[31–35,64], or that relatively steep backset bedforms are specifically a record of the formation and propagation of Froude jumps, where flow transforms from Froude supercritical (>1) to Froude subcritical, similar to fluvial chute-and-pool structures[20,31,35,37,39,64–66] (Fig. 1a/e, f). Our experimental currents show rapidly evolving Froude numbers (Fig. 4). Within the current body, planar beds are deposited at $Fr$ 3–5, shallow backset bedforms at $Fr$ 2–3 and steep backset bedforms at $Fr$ 0.59–2. We show that an apparent Froude jump within the flow forms in the current during deposition of the steep backset bedforms (Fig. 7). As the experimental current is granular, we adopt the term granular jump[67–69], which shares many characteristics with its hydraulic counterpart. However, the outgoing current only briefly has $Fr < 1$, due to thickening of the current directly prior to its

being blocked, meaning that a granular jump, strictly defined as a flow transitioning from $Fr > 1$ to $Fr < 1$, exists here for only 0.1–0.2 s.

As the sediment deposit grows in thickness, a critical point is reached where the incoming flow cannot surpass the positive slope, and the pseudo-jump propagates upstream as a granular bore[68], which travels at 0.14 ms$^{-1}$ between 96 cm and 90 cm along the flume length. Here, we use granular bore to describe the upstream propagation of the depositional front of the granular material, regardless of flow conditions. This process appears to be similar to the stoss-side blocking or granular jamming invoked to explain stoss-aggrading bedforms at Tungurahua[22,70], where the granular current is simply blocked by topography with no particular fluid conditions necessary.

An interesting feature seen in the granular jump of Boudet et al.[67] and our own currents is the steepening of stoss faces well beyond the repose angle at the front of the granular bore, and its collapse by avalanching (Fig. 7d). This is likely caused by rapid deposition from the incoming flow countering the effects of

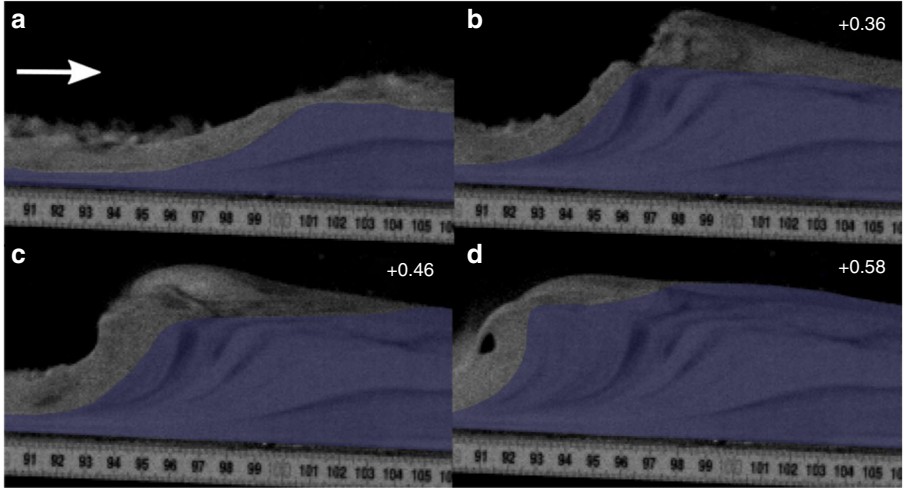

**Fig. 6 Field photos and data of the Pozzolane Rosse ignimbrite erupted from Colli Albani, Italy.** The ruler is 1 m in length. Coordinates are for UTM 33T grid, using the WGS84 Datum. **a** steep stoss-side backset bedform at 323348 4639535, c.f. Fig. 2a–c. **b** Stoss and lee angles for PR and experimental backset bedforms. Several of these backset bedforms have similar stoss angles to our experimental features; however the lee angles are much steeper. **c** backset bedform directly upstream from **a**, c.f. Fig. 2d. **d** Shallow bedform at 323037 4639270, thicker by ~15 cm over the stoss and crest compared with the lee.

**Fig. 7 The formation and evolution of a granular bore.** Numbers in the top right are seconds passed since the first frame. Shaded area shows stationary deposit. Flow direction left to right. **a** shows the initial formation of a steepening bump, with the incoming and outgoing current both supercritical. **b** shows the upstream propagation and further steepening of the bore, immediately after blocking of the outgoing current. **c** The bore propagates further upstream, the front steepening to vertical. **d** The front of the bore collapses upstream by avalanching.

gravity sliding, and allowing the bedforms to steepen well beyond repose angle. Again, a similar phenomenon of very high sedimentation rates is used to explain near-vertical bedding at Tungurahua[70]. The particles deposited by the current as the deposit front steepens form our steep backset bedforms, with stoss angles up to 90°. This may explain why the smallest aeration drop in our experiments (0.66 $U_{mf}$ to 0.53 $U_{mf}$) did not form steep backset bedforms—the drop was too small to promote the levels of deaeration and deceleration necessary for such rapid sedimentation. Our experimental data therefore call the widespread interpretation of backset bedforms recording Froude jumps within dilute PDCs into question, as we show that similar features can form in dense granular flows in relation to an extremely transient Froude jump, and more clearly related to stoss-side blocking.

Calculated $N_S$ and $N_B$ numbers indicate that planar bedsets are deposited under conditions closer to a collision-dominated flow regime ($N_S > 0.1$ and $N_B > 450$[71]) than the backset bedforms (Supplementary Table 1). The planar bedset deposition occurs beyond the transition to the unfluidised section of flume, and therefore they are deposited by a current which is experiencing more collisions between particles due to the loss of gas pore pressure. The backset bedforms are deposited closer to this transition point, where the current has a higher gas pore pressure and grain collisions are not as prevalent. A ratio of $N_B$ to $N_S$ ($N_F$) shows that frictional stresses are considerably higher than viscous shear stresses in the area of the currents depositing steep backset bedforms (Fig. 4d). As the current is waning at this point and relatively thick, this could result in sustained contacts between particles despite relatively high gas pore pressures.

The PR ignimbrite is generally massive and fines poor, which suggests that the flow-boundary zone conditions of the parent PDC were highly concentrated, likely close to the fluid escape-dominated and granular flow-dominated end-members of Branney and Kokelaar[9]. In addition, the dense nature of the clasts, lack of fines and the lack of widespread stratification all suggest that the ignimbrite is the deposit of a dense, granular PDC. The presence of backset bedforms within the deposit, which are typically indicative of dilute, turbulent flow (pyroclastic surges), is therefore paradoxical. Rather, the backset bedforms must have been produced by some other process than turbulence within a dilute current.

The similarities between the structures in the PR ignimbrite and our experimental deposits formed by a dense granular current suggest that the depositional processes involved in both cases could be related. We interpret the undulated bedding facies—which includes the backset bedforms—to have been deposited by the same PDC as the rest of the PR ignimbrite. This is due to the traceable lateral transition between facies, the similarity between the grain-size curves over a range of localities, and because the tephra is compositionally identical in the two lithofacies. Instead, the change in facies could be due to the onset of rapid deposition and stoss-side blocking related to the run-up of the PDC into the Apennine mountains (Fig. 5a). Giordano and Doronzo[62] interpret the undulated bedding to the east of the volcano as the result of rapid sedimentation and a reduction in the lateral mass discharge rate caused by a palaeovalley perpendicular to flow. Our experimental steep stoss-sided bedforms are created in a waning flow regime after the cessation of basal gas injection and the resulting decrease in pore pressure results in rapid sedimentation, so these interpretations are consistent.

We propose a depositional model whereby shallow backset bedforms are deposited by supercritical flow, forming a topographic irregularity which slows the incoming current (Fig. 8a–b), causing stoss-side blocking, forming a granular bore and promoting rapid deposition (Fig. 8c). Continued deposition steepens the front of the bore until it collapses upstream through

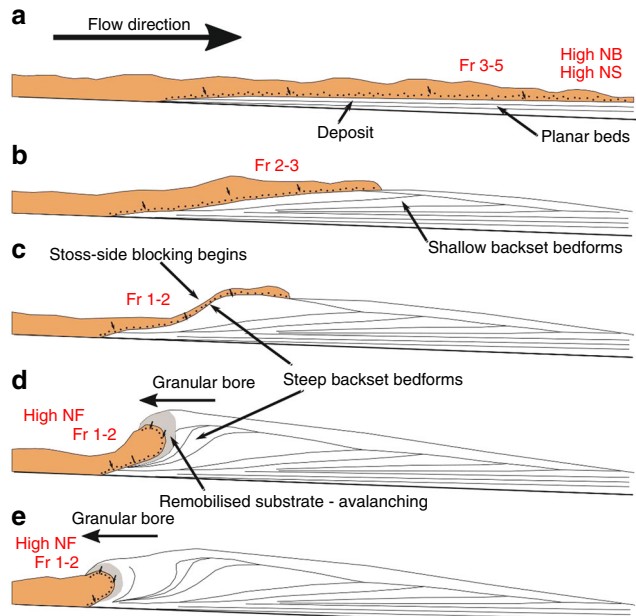

**Fig. 8 Schematic showing how different backset bedforms could be deposited by a PDC.** Flow properties in red (*Fr*, $N_S$, $N_B$, $N_F$) refer to the Froude, Savage, Bagnold and Friction numbers, respectively. See text for detailed description.

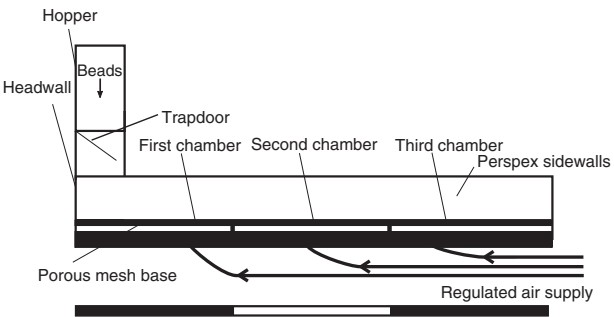

**Fig. 9 A longitudinal section view of the experimental flume.** Scale bar = 3 m.

avalanching (Fig. 8d–e). Our work provides direct evidence that bedforms can be created by dense granular PDCs, and supports the stoss-side blocking process first suggested by Douillet[22,70] based on field deposits.

The upstream propagation of a granular bore, which is caused by the blocking of the current by the aggrading deposit, is a process which in nature could be exacerbated or triggered by pre-existing topography[69]. The waning nature of the incoming flow at this point, and its relatively low Froude number, suggests that while most of these steep backset bedforms are technically recording the transition from supercritical to subcritical flow, both the shallow backset bedforms and planar beds are formed under increasingly supercritical conditions. It follows that shallow backset bedforms and planar bedsets may then be better indicators of supercritical flow conditions when interpreting dense PDC deposits. The proposed phase diagrams presented here are a major step towards quantitative links between PDC processes and their deposits.

Bedforms can be the product of a dense granular flow and can form without any interference (e.g., tractional shear) from an overlying dilute turbulent layer. As the presence of bedforms

(e.g., cross-stratification and backsets) has been commonly used as diagnostic evidence for dilute, turbulent currents, our findings have important implications for field interpretation—as different types of PDCs can react differently to topography the correct classification is necessary for hazard assessment. Other sedimentary characteristics such as field relations, grain size and sorting must be used in order to distinguish between the two PDC end-members. This challenge to the interpretation of the deposits of particulate granular currents is particularly relevant to other free-surface granular mass flows, including landslides, snow avalanches and debris flows. Our experiments demonstrate that formation of different bedforms may by controlled by current thickness and current velocity which has important implications for hazard mapping, and the potential for further investigation to (a) expand the bedform stability criteria identified here, and (b) define palaeoflow conditions from recorded bedforms.

## Methods

**Flume set-up.** We use the experimental flume of Smith et al.[52], modified so that release of the particulate density current is controlled by a trapdoor instead of a horizontal lock gate (Fig. 9), such that colour stratification in the starting charge transmits to the flow and deposit. The base of the flume comprises 1-m-long sections that can provide independently controlled gas fluxes through a porous baseplate in each section in order to fluidise any overpassing material. The flume was kept at an angle of 2°, to promote flow away from the impingement surface while maintaining a sub-horizontal surface.

The air-supply plumbing allows a gas flux to be fed through the base of the flume, producing sustained aeration of the current. In such thin (<0.03 m), rapidly degassing laboratory currents, this enables us to simulate the long-lived high gas pore pressures that characterise thicker PDCs[44,52]. The gas flux supplied through the base in each of the three sections of the channel was controlled to vary the aeration state of the currents, all of which were below minimum fluidisation velocity ($U_{mf}$), as complete fluidisation would result in non-deposition[44].

Various aeration states were used to trigger different flow behaviours. The first chamber (0.66–0.93 $U_{mf}$) always had higher gas flux than the second chamber (0–0.66 $U_{mf}$) to trigger deposition in the target area of the flume. The experiments were recorded using a high-speed camera at 200 frames per second. This video recorded a side-wall area of the channel at 1-m runout (across the contact between the first and second gas-supply chambers), allowing for measurement of the flow conditions. From the opening of the trapdoor to the cessation of deposition, each experimental run lasted approximately 4 s.

**Experimental material and deposits.** The experiments were performed using particles of spherical soda lime ballotini with grain sizes of 45–90 μm (average $D_{32}$ = 63.4 μm calculated from six samples across the material batch) similar to the particles used in previous experimental granular currents[40,42,44]. These ballotini belong to the Group A classification of Geldart[72], comprising particles 45–90 μm, which expand homogenously above $U_{mf}$ until bubbles form, and which are non-cohesive. As PDCs contain dominantly Group A particles, this allows dynamic similarity between the natural and experimental currents[13]. Detailed mechanical properties of the ballotini are presented in Supplementary Table 3, derived from rotating drum[73] and shearbox (BS 1377-7:1990) testing. These give cohesion values of 0 kPa, and an internal friction angle of 25.3° (Supplementary Fig. 3). Static minimum ($\theta_{Smin}$), maximum ($\theta_{Smax}$) and dynamic ($\theta_{Dyn}$) angles of repose are found to be of 11.7°, 31.9° and 20.9°, respectively (Supplementary Fig. 2).

Due to the monodisperse nature of the materials, any internal structure is easily masked by lack of contrast between packages of sediment[74]. To this end, the charge for each experiment was built up of layers of dyed beads so that flow packages could be tracked throughout flow and deposition, as used in Rowley et al.[44]. Reported velocities are calculated by tracking these coloured sediment packages in the body of the current immediately prior to their deposition.

When reporting the length of a bedform, the distance from the onset of the stoss-side lamina to the termination of the lee slope on the depositional surface was measured. Thickness refers to the distance between the lowest point of a lamina in the bedform to the highest point of a lamina in that same bedform (Fig. 1g, h). Bedform lengths and thicknesses are reported, as opposed to wavelengths and amplitudes, as we do not produce repetitive trains of bedforms. This is because of the short nature of the experiments—the current is not sustained for long enough, and doing so would require an unfeasible amount of material under the current set-up.

**Error measurements.** Errors (2 s.d.) for various measurements are as follows: current thickness: ±0.0013 m. Current velocity: ±0.055 ms⁻¹. *Fr*: ±0.17. $N_F$: ±67,000.

## Data availability

Data supporting the graphs in Fig. 4 are derived from raw video files, and are available in Supplementary Table 1. One experimental run is available as Supplementary Movie 1. Four other videos are available upon reasonable request.

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

## Acknowledgements

This work was carried out as part of a PhD project funded by a University of Hull PhD scholarship in the Catastrophic Flows Research Cluster. Experiments were performed in the Geohazards Lab at the University of Portsmouth, using equipment funded by a British Society for Geomorphology Early Career Researcher Grant held by P.R. D.P. was supported through funding from the European Research Council (ERC) under the European Union's Horizon 2020 Research and Innovation Programme (Grant Agreement no. 72955). G.G., M.T. and A.S. gratefully acknowledge The Grant of Excellence Departments, MIUR-Italy.

## Author contributions

G.S. carried out experimental work and drafted the paper. G.S., P.R., G.G., M.T. and A.S. carried out fieldwork. G.S., P.R and R.W. analysed the experimental data. G.S., P.R., R.W., G.G., M.T., A.S. and D.P. discussed results and edited/commented on the paper. Characterisation of the experimental materials was led by S.C.

## Competing interests

The authors declare no competing interests.
