## [Peer Review File · Nature Communications]

Reviewers' comments:

Reviewer #1 (Remarks to the Author):

Review of Nature Communications manuscript:

A bedform phase diagram for dense granular currents

Authors: Smith, Rowley, Williams, Giordano, Trolese, Silleni, Parsons, and Capon

Reviewer: Benjamin Andrews, Smithsonian Global Volcanism Program

Smith et al. present experimental results that show how certain depositional structures generally attributed to transport and deposition from turbulent particle-laden density currents can be formed from granular flows. Specifically, the authors show that regressive dune bedforms (that is, upstream propagating dune structures) can be formed by an upstream propagating granular bore under waning flow conditions. The conditions at which this process occurs are predictable based upon the changes in the flow's Froude and/or Friction numbers. The authors use these insights to suggest that portions of the Pozzolane Rosse ignimbrite featuring steep regressive dune bedforms were deposited by granular flows (not by dilute surges).

The implications of this study are very interesting. As Smith et al. point out, much of our knowledge of PDCs (and thus our ability to forecast their behavior) is based upon interpretation of PDC deposits. Much of our understanding of the processes that form deposits is, however, based upon deposition in fluvial systems; studies in recent years have shown that the complex multi-phase and multi-regime currents that are PDCs exhibit transport and depositional behaviors that are not always represented in fluvial systems. To that end, this study is a strong contribution towards elucidating how PDC deposits can form. Indeed, the paper is nicely focused and provides a good case study of how bedforms that might be attributed to turbulent or tractional processes can instead record a purely granular process. In addition, the authors present their findings in the context of simple scaling (this is good and elegant!) that shows how laboratory experiments can apply to natural currents 1000s of times larger.

The paper could be strengthened with a few minor revisions. First, the methods section should explicitly describe the experimental durations, and the authors should mention the upstream propagation velocity of the granular bore; together, this will allow the reader to assess how steadiness/unsteadiness/impulsiveness of the current affects the bedforms (this will also allow for a richer interpretation of natural PDCs and the time recorded by individual depositional units). Second, the authors focus on regressive bedforms, which is appropriate, given that is what their experiments produced, but a brief mention (even 1 sentence) in the discussion about whether or not granular flows can form progressive bedforms would be welcome.

In summary, this is a well-written paper. Smith et al. clearly present experimental results and discuss their findings in the context of a natural ignimbrite deposit. The study has important implications for the evaluation (or re-evaluation and interpretation) of pyroclastic density current deposits around the world. I recommend publication with very minor revision.

Benjamin J. Andrews

Smithsonian Global Volcanism Program

Specific comments:

Line 47 – the second “PDCs” should have a possessive apostrophe, “PDC’s”

Lines 91-95 – were there no progressive dune bedforms observed in your experiments? If so, please state that clearly in this paragraph.

Eq. 2 and Eq. 3 – this is very minor, but please put the terms in the numerators of the equations in the same order – this makes it easier to see that NB and NS have the same numerators.

Fig. 4 – please label the panels a, b, c, d.

Reviewer #2 (Remarks to the Author):

Dear authors and editor,

I have read with attention the manuscript by Smith et al. entitled "A bedform phase diagram for dense granular currents" submitted to Nature.

-I declare here that the main author sent me spontaneously the submitted version of his manuscript by email, after we met during EGU2019 and I had asked for news. I do not consider this creating any conflict of interest and feel perfectly safe about providing a review.-

This is a nice experimental study with fluidized granular flows that show how concentrated PDCs could form backset beddings with cross-lineations or cross-stratifications. It shows how a kind of bore is triggered by the forming bedform and becomes unstably steepening. The experimental results are then compared to field examples, and a phase diagram based on the experimental results is suggested.

In brief, the introduction is in general clear and understandable. The experiments are amazing and novel. I agree with the interpretation, and the data is strengthening many ideas that I raised in several field based manuscripts about dune bedforms forming from dense granular currents (see main comment 2 and line by line 209-213). Something I'd like to suggest to discuss or take into account is the possibility of a "granular jamming" in these experiments (it seems described but not called).

Some problems with the writing/structure can be corrected quite easily (yet still involving a bit of work). In particular, I think it really needs a better transition at the end of introduction. I would also put more accent on the bed-aeration story /fluidized-currents. I think the part on the field examples can be significantly shortened to focus on the main story delivered by the experiments. I have two bigger negative comment:

1. The manuscript makes a confusion between cross-laminations (i.e. mm scale laminae that are not planar) and dune bedforms (anything above the m-scale that shows a non planar surface or paleo-surface). It then states that the presence of regressive bedforms is accepted as proving parental dilute PDCs. This is, to my opinion, a strong amalgamation of ideas and wrong. Of course, regressive bedforms can form from concentrated granular PDCs, and several studies have pointed that out (Brown and Branney 2004, Douillet et al. 2013b, 2018b, see main comment 2). I disagree that there is a novelty in that part.

2. I am really unsatisfied by the field pictures, which I don't find are serving as it should the manuscript.

All in all, this is a nice study and the comments can be adapted relatively easily (involving quite a bit of time though). Please find below my 6 main comments, and then more punctual line-by-line comments. I'm honestly not sure that anyone cares about dune bedforms in pyroclastic currents apart from me and the authors (but this is only a depressed late Friday thought...), so who knows how this study will be received by the broad audience of Nature... I'd vote to give it a chance.

Regards,
Guilhem Amin Douillet

The rest of the comments are available from the attached pdf file

Reviewer #3 (Remarks to the Author):

This is an interesting paper that makes a decent contribution to understanding of bedforms in an experimental setup and some possible implications for pyroclastic density currents.

However at present there are three major problems with this paper which in my opinion must be addressed fully before it can be considered for publication:

1. More needs to be done to justify why these experiments are analogous to hot, gas-driven PDCs. What about scaling between the experiments and natural PDCs? Also what about the grain size differences. These are clearly mentioned and that there are important differences but not what the implications are. Also there is mention that trains of bedforms are not produced (that are seen in reality). Why not? These kind of things – grain size differences, bedform differences, all should be discussed more so that the drawbacks and limitations of the experiments are clear when comparing to reality.

2. The material on the 'Field Analogue' - the Pozzolane Rosse ignimbrite is far too cursory to support what is being proposed here. It would take a considerable amount of additional text and possibly figures too to justify the 'bedforms in ignimbrite' idea. The reader has to be absolutely convinced that these are not related to a pulse of the eruption with quite different physical characteristics. It is essential to demonstrate that these bedforms are intrinsically related to the massive deposits.

One solution which could address this would be the removal of the 'Field analogue' part. I think that the paper makes a large enough contribution without the Field Analogue part and a simple option would be to remove that and just focus on the bedforms phase diagrams idea. Which is what the paper title is. At present it appears that the Field Analogue part is just an afterthought.

3. There are many places in the paper where it would benefit enormously from more detail and/or more extensive discussion (as indicated below in specific points). At present many points, issues etc don't get the discussion they need to and many of the arguments end up appearing cursory and not fully discussed.

Specific problems

Line 60 Ignimbrite – a widely used term in volcanology but it needs to be defined carefully here. Some of the references given here are for bedforms generated in deposits which would not be defined as ignimbrite by many volcanologists. For example I don't think many volcanologists would term thinly bedded PDC deposits associated with small tuff cones as ignimbrite. So for this reason it is essential to define exactly what you mean by ignimbrite.

Line 71 – The regressive / progressive subdivision of bedforms comes from the sedimentology literature. Allen (1984) should be cited here. Even in the volcanological literature it didn't start with Brand et al 2016. So I think that the literature can be given a bit more acknowledgement here.

Line 81 – Quasi steady currents - this is proposed without any justification as far as I can see. Perhaps I am missing something in the odd setup of the journal with methods etc given later, but if there is further information please give it, as at present the reader is just left wondering.

Line 84 – don't think its necessary to say 'for the first time' again here. You said that 8 lines back.

Line 99 – Perhaps a few more lines on what the drawbacks might be of not producing 'trains of bedforms', why are these not produced when they are in nature? This could be developed in the discussion a bit more.

Line 108 – stating > 20o is certainly an over generalisation, some are 90o and some look almost 'overturned'

Line 111 – I think there could be a sentence or two of introduction here. It launches straight in.

Line 114 – '...more rapid deaeration' another term dropped in without any info. The reader is kind of left high and dry...More explanation is needed.

Line 162 – Field Analogue

The first sentence of this section is a bold unsubstantiated statement and is unacceptable. The field evidence needs to be presented first, then an interpretation arrived at. The reader needs much more descriptive information. The interpretation of these type of deposits is just that an interpretation and we need more than 'matrix supported and fines-depleted' to reach a reasoned conclusion. Its absolutely vital to the understanding. Fines-depleted is a poor term (it also implies an interpretation, how do you know there was fines in the first place?). Why not use 'fines poor' which is a descriptive term.

Line 162 Delete 'and in places partially lithified by zeolites' . It's irrelevant to the story here.

Line 169 Grainsize of 6 samples..... how do we know these are representative of anything? There is no information on the location, facies, stratigraphic position as far as I can see.

Line 176 -179 So clearly these natural deposits have different properties to the experimental setup. Is this not a serious drawback? This must have some further discussion somewhere. Owing to its importance in terms of the differences between experiments and field.

Line 180 – but you state that it isn't a massive ignimbrite.....How the massive parts relate to the dune bedforms is absolutely critical. At present this information is absent

Line 180 – 185 This paragraph is just far too cursory to address this contentious and complex issue. It absolutely cannot be dealt with in 5.5 lines. What is provided is nowhere nearly enough information.

Line 187 and Figures 7a and c . These photographs are not clear enough in my opinion, they need to be much clearer. In particular 7c could be anything. The upstream beds are much steeper than 20o. So are your experimental ones. Why not be more specific and give actual angles ?

Line 191 is there a reference for these other bedforms that are found in the Pozzolane Rosse? If so please cite it here.

Line 215 In the photographs some parts look like they are 'overturned' in places (Fig 9 d and E). I'm not sure I have seen this in Field analogues. Perhaps you can expand on the similarities and differences more here ? Give more references to those field examples which show these similarities perhaps?

Line 228 The field analogue certainly doesn't support this 'experimental supposition that dune bedforms can be formed by dense granular currents. It could do if there was adequate careful description and interpretation of what is present in the field, rather than a couple of overly cursory sentences.

Line 251 – generally apart from where the term regressive is introduced there are no progressive bedforms mentioned. Presumably as they were not generated in the experiments however perhaps there could be more discussion about why you are not seeing/generating progressive bedforms in these experiments and the significance of this.

Line 264 - I think the suggestion that dune bedforms can be formed by dense flows is

demonstrated by this paper. However there are such a wide range of bedforms in reality, in terms of size, wavelength etc that more than just grainsize characteristics are needed. Detailed field relations must be the most important feature to distinguish the two different end members.

We thank the reviewers for their comments and have revised our manuscript accordingly. The resubmitted manuscript is a significantly improved piece of work. Below we detail the changes we have made to reviewers' comments. Line numbers in reviewers comments refer to the original manuscript, our response in bold refers to the revised manuscript.

Reviewer 1 – Ben Andrews

The paper could be strengthened with a few minor revisions. First, the methods section should explicitly describe the experimental durations,

Added duration to the end of the methods (line 336-337).

and the authors should mention the upstream propagation velocity of the granular bore; together, this will allow the reader to assess how steadiness/unsteadiness/impulsiveness of the current affects the bedforms (this will also allow for a richer interpretation of natural PDCs and the time recorded by individual depositional units).

Granular bore velocity added (line 233).

Second, the authors focus on regressive bedforms, which is appropriate, given that is what their experiments produced, but a brief mention (even 1 sentence) in the discussion about whether or not granular flows can form progressive bedforms would be welcome.

Sentence about absence of progressive bedforms added in the discussion (Line 216-217).

Specific comments:

Line 47 – the second “PDCs” should have a possessive apostrophe, “PDC’s”

Replaced with ‘their’ (line 49).

Lines 91-95 – were there no progressive dune bedforms observed in your experiments? If so, please state that clearly in this paragraph.

Text stating that no progressive bedforms were observed added (line 108)

Eq. 2 and Eq. 3 – this is very minor, but please put the terms in the numerators of the equations in the same order – this makes it easier to see that NB and NS have the same numerators.

Corrected (line 155-156).

Fig. 4 – please label the panels a, b, c, d.

Corrected.

Reviewer 2 – Guilhem Douillet

Some problems with the writing/structure can be corrected quite easily (yet still involving a bit of work). In particular, I think it really needs a better transition at the end of introduction. I would also put more accent on the

bed-aeration story /fluidized-currents. I think the part on the field examples can be significantly shortened to focus on the main story delivered by the experiments. I have two bigger negative comment:

1. The manuscript makes a confusion between cross-laminations (i.e. mm scale laminae that are not planar) and dune bedforms (anything above the m-scale that shows a non planar surface or paleo-surface). It then states that the presence of regressive bedforms is accepted as proving parental dilute PDCs. This is, to my opinion, a strong amalgamation of ideas and wrong. Of course, regressive bedforms can form from concentrated granular PDCs, and several studies have pointed that out (Brown and Branney 2004, Douillet et al. 2013b, 2018b, see main comment 2). I disagree that there is a novelty in that part.

We have decided that our experimental structures are best described as ‘backset bedforms’ rather than ‘dune bedforms’, in agreement with your point about scaling. We have mentioned that regressive bedforms being formed by supercritical flow in dilute PDCs was the original interpretation and is still widespread in the literature (line 218-222) and now do not state that our work is the first to prove that these bedforms can be formed from dense PDCs.

2. I am really unsatisfied by the field pictures, which I don't find are serving as it should the manuscript.

We have taken the editors advice to continue with the inclusion of the fieldwork aspect of this paper, but appreciate reviewers feedback on the quality and clarity of the field photos. New field photos have been added to fig. 7 with explanatory text (line 196-205), and interpretive guidelines have been illustrated onto the image to help the reader in observing the features discussed.

Main comments: 1 The lacking pivot point between introduction-result: A pivot point that seems to lack any transition, or sufficient explanation is at line 88. I find that there is a problem with the structure here, I guess it's the Nature format that does not allow the "Methodology chapter" to be after the introduction, although it was here in a previous version. This is strongly felt. In any case, I recommend to clearly state here that your setup is made of a first part where the mixture is fluidized by injection of air from the bed, and a second part, where the artificial injection of air has ceased or is decreased and deposition occurs. Your lines 241-243 could actually fit well here (maybe even 241-250). This is what I understood from your presentations at conferences, and feels necessary to understand the results.

We have modified the end of the introduction. Extra methodology information has been added from the discussion so that the reader can more readily understand the experimental procedure without resorting to the methods (line 90-97).

2 Field example: 2a. I barely see anything in the field pictures, I think it is not understandable for the general reader (this is the softest version I can find). I'd really urge to get better field data. Apologies for the self advertisement, but your experiments really ring a bell in what I saw in natural examples, and referring to them might really strengthen your point (although it would remove part of the novelty, but the experiments are really amazing enough to not need this). Here a few examples: -I honestly believe that a natural upscaling of your experimental steep structures could be what I observed at Tungurahua (Figure 16 and 17 in Douillet et al. 2018b), which are also interpreted as dense flows: "The parental currents were likely to be dominated by particle–particle interactions during their transportational phase in order to support the coarse content of the deposits, a hypothesis reinforced by the bad sorting of this facies". I also believe that we came to the same interpretation for the formation of these massive backset beds, in what I called "granular jamming" (see your line 209 vs. fig. 18E in

Douillet et al 2018b). I think we also agree on the oversteepening being due to very rapid backset aggradation (see in Douillet et al. 2018b: "the organization of the filling as subvertical aggrading lamination [...] is not likely to be preserved without extremely fast burial to avoid gravitational collapse. Thus, very high sedimentation rates [...] occurred within seconds within a pressing/plastering current." Also, your experimental results are, albeit scale, exactly identical to the coarse dune bedforms at Laacher See (Figure 8a-8b in Douillet et al. 2018a). -For your shallow regressive bedform, they are extremely similar to the elongate structures in Douillet et al. 2013b (Figure 4). Note, these are interpreted as related to dense flows, and as you did here, I stated that "Rather than assign these as antidunes, we suggest that the bedslope decrease induced on the stoss side of the bedform would lead to blocking of parts of the bedload causing stoss- aggradation without the flow necessarily being in a trans- or supercritical condition". -For a natural full scaled example of your experiments, I am convinced that you are modeling the backset ripples from Tungurahua (Figure 7 of Douillet et al. 2018b). -Not sure I can suggest this as a reviewer, but please do not hesitate to come to me if you wish some eloquent field pictures for a future version...

We have added new annotated field photos. We also agree that the Tungurahua structures are valid analogues and have cited Douillet et al. 2013 and Douillet et al. 2018 where appropriate. See specific comments below for our agreement with the 'granular jamming' phenomenon.

- 2b: Another point concerning your field example but I am not sure about this: May these high emplacement temperatures and the presence of lithified zeolites be a sign of sticky particles at time of deposition? Some colleagues might say that this is the reason why you see backset lineations in those deposits rather than a pure granular flow process.

This is an interesting point, but outside the scope of this study. Without high temperature experiments on these particles this hypothesis is untestable, and such a high temperature flume or shear cell apparatus is currently unavailable. We don't see any evidence for sintering or welding of particles in the field, so there is a lack of evidence for the Pozzalone Rosse PDC being hot enough to have 'sticky particles' at the time of deposition.

2c: I don't think that the grain size data are necessary here. The material is coarse grained, quite massive, and generally interpreted as dense PDC, this is sufficient for your aim. No need to add grain size data, at least in the main text and for the brief format of Nature articles.

In meeting with reviewer three's wishes for more detailed field descriptions we would prefer to keep the grain size data, although we have changed its presentation (fig. 5).

3 "Dune bedform" nomenclature: You produce ripple-sized structures, why calling these dune bedforms (dunes are notoriously larger than 0.2 m). What about "backset ripples" (that's how I called similar examples from the field in Douillet et al. 2018b). The upscaling to dune bedforms is not straightforward to me and already interpretative. Maybe "backset structures" if you do not wish to use a nomenclature I already introduced? In any case, the size is not the one of dune bedforms in the experiments...

We agree that 'dune bedform' may not be the most appropriate term. To keep things as descriptive as possible we have opted to use 'backset bedform' instead.

4 About the experiments and the aeration story: 4a. From what I understand from figure 2 and its legend, it's the abrupt change in aeration that leads to the 3 possible scenarios, not necessarily the flow velocity/thickness during deposition. - What is said in Figure 2 is not frankly visible in the results, where accent is put on the Froude/ friction numbers...-

Extra text about the effects aeration has been added in the results (line 142-145) and discussion (line 246-248).

4b. I am similarly wondering if these structures are not the result of a granular jamming effect due to the growing morphology (see also other comments along the text)? I'd be very interested to see what comes out by plotting "deposit thickness" vs "flow thickness" by type...

We agree that 'granular jamming' (Douillet et al. 2018) is the mechanism we are modelling, however, we prefer the descriptive term 'stoss-side blocking' (Douillet et al. 2013), as granular jamming has a specific meaning in the field of granular physics (e.g. Silbert et al. 2002, Bandi et al. 2013, Jiang et al. 2014).

4c. Also, it seems that all your currents are ending at the bedform, none is overpassing the structures produced. Does that fit with your view of natural currents? Could you discuss this?

Overpassing does occur until the deposit is high enough to entirely stop the incoming current through stoss-side blocking (line 231-232).

4d. Related comment: all structures presented here are regressive. I think this is noteworthy and could be a strong bias... Could you also reproduce any structure with a downstream aggrading trend? This should/would be worth mentioning.

Text added about absence of progressive bedforms (line 216-217).

5 Focusing the story: You are writing about experiments that create regressive bedforms from dense granular flows. The background of Pouzzolane is not the subject, so leave it away. I'd cut at least by half the part from 161-192. At least lines 164-166, 173-174 186-187 are unnecessary, and you could make short cuts from observations to interpretation of the parent granular flow (at least in a Nature-type manuscript). An example to shortcut by half the words on lines 186-192, is given in the line by line comments.

This entire section has been rewritten as per the wishes of reviewer three, and we have erred on the side of more detail.

6 The "pseudo summary-conclusion": I have the feeling that the end from lines 241-272 is supposed to be a conclusion, but is not called as such. I would either put a title chapter such as conclusion or summary, or shorten these 30 lines in 3 sentences in a single chapter. It feels very repetitive as it is, and I am wondering if I missed something...

We have rewritten this section to shorten and make it less repetitive as suggested (line 297-314).

Line by line comments:

39: not sure "biggest" is appropriate, I suggest "largest", even "one of the largest" maybe

Changed to "largest" (line 41).

43-44: the sentence on hazard is, to my opinion, out of subject and unnecessary.

We disagree, ultimately, our research group is motivated to study ignimbrite emplacement as a mechanism to better understand PDC behaviour and to better improve our interpretation of deposits to improve our hazard assessments. As the driver of our study, we would rather keep this in.

49-50: I'd rather put interpretation of the geological first and as a second part, the analogue and

numerical modeling (I find this chronologically and in terms of thinking more appropriate)

Changed as suggested (line 52-53).

52-56: I find necessary to briefly explain what is a dense PDC before using that term. This is what is done next paragraph, so you could just shift paragraph 52-56 after 57-63. You might find interesting references in the introduction of Douillet et al. 2013a concerning the end members of PDCs...

Shifted paragraph about analogue modelling of dense PDCs to lines 78-82, seems to fit better there. Added two more references in PDC end member section (Wohletz & Sheridan 1979 and Walker 1984, line 57).

57-59: references 27-31 are all unrelated to volcanoes! (I think you could cite Branney and Kokelaar 2002 and Douillet et al 2013a, 2013b here).

These references are purposefully not about volcanoes to show how the study of these features in PDC deposits grew out of the sedimentary literature.

64: I suggest to be more precise with this "include structures whose morphologies have drawn comparisons" and replace by "include backset laminations which have drawn"? or "include stoss-side aggradation as is observed for"?

Replaced with "formed by stoss-side aggradation, similar to..." (line 61-62).

68 to 69: I think there is no need for a new paragraph here, is there?

Changed as suggested.

72: I'd be delighted if you could add "unrelated to supercritical flow conditions" about the regressive climbing dunes, please.

The point of this paragraph is to clarify descriptive terminology so we would prefer to leave interpretative statements out.

77-88: I think this paragraph should be reworked and could have a much stronger impact and appear clearer. (see main comment 1)

Extra methodology information has been added from the discussion so that the reader can more readily understand the experimental procedure without resorting to the methods (line 90-97).

78-80: I have the feeling that the sentence is not grammatically correct or lacks a word but I am not a native speaker...

Added "we" on line 85.

97: unclear what "packages" and "beds" are. Do you mean laminasets/bedsets/layers? Maybe

the nomenclature used in Douillet et al. 2013b could be useful here.

Agreed the Douillet et al. 2013b terminology is useful, have used “lamina” and “bedsets” to replace package/bed throughout the manuscript.

98-102: These sentences belong to your method rather than to results.

Moved to methods (line 355-359).

101: not sure it is a half wavelength, it does seem a full one to me...

We have removed the half wavelength reference – as no trains of bedforms are formed this can't be properly measured.

102: unclear, does it mean from the lowest point of a lamina to the highest point, or the full thickness of the final deposit, in which case that has no real physical meaning for the bedform...

Thickness refers to the distance between the lowest point of a lamina in that bedform to the highest point of a lamina in that bedform. Sentence rewritten to clarify this (line 356-358).

103-110: I really recommend to switch those lengthy sentences only aiming at giving numbers into a table. I find that it breaks the flow and is not very pleasant to read.

Modified text into a table (line 116).

"Dune bedform deposition": this chapter is quite unclear to me for several reasons...

115: "velocities of ~1-1.5 ms⁻¹"... where and when is this velocity coming from? before or after the break of aeration? at which distance from the break? During the whole duration of the flow?

116: "As the current velocity further decreases and deposit thickness increases" where or when is that? velocity decreases downstream or with time? deposit thickness increases as long as you have net aggradation, so do you solely mean that time passes or is the rate of deposition increasing?

118: "At lower velocities still" you mean at the latest moments of the flow? or the downstream part? or for experiments involving slow flows?

...This is honestly quite unclear, and I am not sure at whole what is meant with those velocities.

We see how the wording throughout this paragraph was confusing, and it has been rewritten for clarity (line 124-131). The point of the paragraph was to show the velocities are decreasing with time.

123: "Current velocities and thicknesses during deposition of the dune bedforms were measured"... You just gave those velocities in the previous paragraph, so this sentence is not useful here. I would either put it earlier or remove it.

Sentence removed; section about supplementary table was added to previous paragraph (line 132).

124: the reader doesn't care about errors in the main text (keep this for the method or supplementary material though). Again, I'd remove numbers in the text and this sentence.

Errors removed to methodology (line 364-365).

131: I would have understood this sentence from the first read if you added a coma after "thickness"... (just giving my non native feeling).

Added comma as suggested (line 139).

132-133: I don't understand this sentence... is it correct? if yes, could you make it simpler for non-native? or should the coma simply be put after "result"?

Added "of" between "result" and "thickening" (line 140)

134: I'd suggest retitling this chapter "Phase diagrams or phase fields" rather than dimensionless Parameters

Retitled "Phase fields" (line 146).

135: please add: the three "TYPES OF" dune bedforms

Changed as suggested (line 147).

136: I am not familiar with the friction number and google didn't help me a lot... I had to read the Iverson 97 paper to find out. The latter citation states that "the ratio of the Bagnold number to Savage number forms a version of the friction number identified by Iverson and LaHusen [1993]". Maybe it would be worth citing this latter reference (I didn't check it further though)?

Reference added as suggested (line 152).

148-150,150-152 and 153 (also 187): I generally dislike the form of sentences of the type "Figure xx shows that" and would rather encourage to directly state what you want, with the call to the figure in brackets at the end. Here: "Different ranges of Fr were measured for the three types of bedforms with greater overlap between the steep and shallow regressive fields than between the shallow regressive and planar bed fields (Fig 4a).

Fixed throughout this paragraph as suggested (line 159-166).

Also, I have the feeling there is sometimes a shortcut between a "dune bedform type" and a "dune bedform", which makes the reading awkward... It seems there are only three structures that were produced, although I believe you have made 100s of dune bedforms that plot in three types of dune bedforms.

Added 'type' or 'types of' throughout the manuscript where appropriate.

154-157: again, listing numbers is not pleasant to read, is it necessary? Could you succinctly explain what it means for the understanding/interpretation rather than actually cite these numbers (they are already visible in the graphs; if needed, put them in a table as a supplementary)?

Rewritten to reduce the amount of numbers listed (line 159-166).

158: I would make this statement a bit less strong by adding some freedom of interpretation, something like: "The planar-shallow-steep sequence of the dune bedform formation **THUS CAN BE SEEN AS** recording the transition..."

Changed to read less strongly (line 167-168).

186-192: I would shortcut by half the words of this part, something like:

"Dune bedforms are found in the Pouzzolane with stoss sides angles greater than 20°, similar to our steep experimental structures (Fig 7a-c). Field observations of shallow stoss-sided dune bedforms have however greater lee than in our experiments, possibly as a consequence of the high repose angle of the natural material (Fig 7d)."

This entire section has been rewritten as per the wishes of reviewer three, and we have erred on the side of more detail.

196: Hydraulic jump is not the right term to my opinion. I use "Froude jump" since a while, and think that would be more appropriate here. Adopting granular jump is great!

Changed occurrences of "hydraulic jump" to "Froude jump" throughout the manuscript.

199: I suggest to replace fluvial counterpart with "fluid" or "liquid"? (hydraulic jumps occur in tidal, turbiditic and many other environments that are not fluvial...)

Replaced with "fluid" throughout the manuscript.

I dont get that sentence "Although the current has a lower Fr immediately after the jump than the incoming current, this is not always <1." If you don't become subcritical, then it's not a hydraulic/Froude-jump, but rather only jamming (again coming to this idea...).

Text rewritten – we've changed our interpretation to say that a brief Froude jump exists before stoss-side blocking occurs (line 228-230).

201: I disagree with the statement that "steep regressive dune bedforms" occur in hydraulic jumps. This is barely seen, and most field as well as experimental studies need extremely high vertical exaggeration to get steep sided backsets... If you want to keep it, please give a few

citations...

This section has been rewritten for clarity, we are saying that such steep backset structures have been interpreted in the literature as recording hydraulic jumps regardless of current-day views (line 218-222).

207: I suggest rephrasing as: "We have THE EXPERIMENTAL DATA demonstrated that this interpretation can be incorrect..."

Rephrased as "Our experimental data call... into question" (line 249-252)

209: I really like this view "As the sediment deposit which triggered the jump grows in thickness, a critical point is reached where the incoming flow cannot surpass the negative slope". I believe that you are actually talking about a granular jamming, as I suggested for backset massive layers forming low-angle elongate beds at Tungurahua (see Figure 18E in Douillet et al. 2018b). It may strengthen your point to refer to this.

Agreed, although as previously mentioned we prefer 'stoss-side blocking' (line 235-238).

213-214: Same, I really like this view: "steepening well beyond the repose angle [...] caused by rapid deposition from the incoming flow countering the effects of gravity sliding". Again, you may refer to Douillet et al. 2018b that similarly state "subvertical aggrading lamination [...] is not likely to be preserved without extremely fast burial to avoid gravitational collapse. Thus, very high sedimentation rates [...] occurred within seconds within a pressing/plastering current."

Agreed, have included reference as suggested (line 243-244).

216: stoss angles up to 90°: please find field examples in Douillet et al. 2018b (figures 5a 5b 5c and chapter "Steep backset lineations")

For clarity, we would prefer to keep specific field examples to our own fieldwork. However, we agree that the Tungurahua examples do provide some nice comparisons and have referred to them throughout this section (line 236, 244).

229: I am quite unhappy with this term of "experimental supposition", and would feel honored if you could mention as well that this was already suggested previously in my field based studies (Douillet et al. 2013b, 2018b)...

We have credited Douillet et al 2013, 2018 as asked (line 289-290).

236: how can a symmetrical bedform migrate upstream? There must be a preferential stossaggradation to migrate upstream, and it is thus not symmetrical... Or it is simply growing equally on both sides, and the crests remains stable and there is no upstream migration? I am generally against the term migration since for PDCs, we generally see preferential aggradation but no real

migration of the whole structure as is the case for aeolian dunes or ripples. I think this is not a small semantic issue to ignore, but something quite fundamental...

Agreed, we have changed 'migrate upstream' and 'upstream migration' to 'stoss-side aggradation' throughout the manuscript.

238: If I understand you correctly here, you suggest that the pool created by the bedform initiation triggers a jamming. I think this is similar to what I suggested in Douillet et al. 2018b for backset lensoidal layers of massive material":they can be understood as the signature of a simple damming triggered in the pools at the toe of bedforms. No particular flow conditions are required and this is understood as a pure topography-triggered jamming or frictional freezing, due to the parental 'granular' part of the flows tripping against the obstacle formed by bedforms." If this is similar, I think that both your study and mine would benefit from a citation here.

Agreed, although as previously mentioned we prefer 'stoss-side blocking' (line 294).

241-250: This sentence would actually fit well at the end of the introduction rather than here. Also it is almost repetitive with the next paragraph starting at 251...

Moved to the introduction (line 90-97).

252-254: It feels repetitive, I think I have already read a lot about this idea higher in the text. Could you actually merge 236-240 with 255 and latter by some rephrasing and leave out 242-250 in the intro and 252-254 out? This part is somehow getting slow and repetitive.

Sentences moved and deleted as suggested.

...Starting from 241, I have the feeling that you are only repeating your results in what could be a conclusion but is not called a conclusion. It seems awkward, and I would either suggest to put a Conclusion title or remove/shorten extremely all this part.

Shortened as suggested (line 297-314).

296:Experimental material: Is the fine-grained material used in this study influenced by Van der Waals forces and other contact forces that might not apply to coarser natural material?

Text added – particles are non-cohesive (line 343).

Figures:

Fig 1: you might like or not to add the graphe from Douillet et al. 2018b with a sketch of a natural bedform (Fig 18a)...

We prefer the diagram from Douillet et al. 2013 as we feel it is more generic.

Fig 4: please put a b c d on the graphs

Corrected.

505: Caption for d: if using the same convention for all (i.e. xaxis vs zaxis), then it is reversed for Friction number vs. Froude number

Changed to be consistent, Froude number always on y-axis.

Fig 5: I think this figure is not necessary and could be in the supplementary material instead

As previously mentioned we would like to present as much field data as possible, although we have changed this figure to include a location map as well.

Fig 6: I am not even sure that this one is really useful, (see my point about grain size). This almost seems a condensation of may data from a previous version...

We have changed this figure to cut down on irrelevant information and display data from the Giordano & Dobran 1994.

Fig 7: I have clearly stated my opinion about your field pictures in main comment 2... I don't even see the pen in fig.7C...

We have presented new field photos in fig 7.

Fig 8: this is nice, but is it possible to add a contour line of the shaded area to see it more precisely?

Replaced cross-hatching with shaded area.

Fig 9: I am worried that the publisher will make this picture a one column width, and that no details will be seen anymore... Could you make a version that is much less wide to be sure this does not happen?

We do not think that fig. 9 could be feasibly made considerably less wide, but leave this decision to the editor.

Reviewer 3

This is an interesting paper that makes a decent contribution to understanding of bedforms in an experimental setup and some possible implications for pyroclastic density currents.

However at present there are three major problems with this paper which I my opinion must be addressed fully before it can be considered for publication:

1. More needs to be done to justify why these experiments are analogous to hot, gas-driven PDCs. What about

scaling between the experiments and natural PDCs? Also what about the grain size differences. These are clearly mentioned and that there are important differences but not what the implications are. Also there is mention that trains of bedforms are not produced (that are seen in reality). Why not? These kind of things – grain size differences, bedform differences, all should be discussed more so that the drawbacks and limitations of the experiments are clear when comparing to reality.

We have further addressed differences between the experiments and reality. Grain differences (line 189-193), lack of trains of bedforms (line 361-364), lack of overturned beds in the field (line 200-203) are all discussed in more detail.

2. The material on the 'Field Analogue' - the Pozzolane Rosse ignimbrite is far too cursory to support what is being proposed here. It would take a considerable amount of additional text and possibly figures too to justify the 'bedforms in ignimbrite' idea. The reader has to be absolutely convinced that these are not related to a pulse of the eruption with quite different physical characteristics. It is essential to demonstrate that these bedforms are intrinsically related to the massive deposits.

One solution which could address this would be the removal of the 'Field analogue' part. I think that the paper makes a large enough contribution without the Field Analogue part and a simple option would be to remove that and just focus on the bedforms phase diagrams idea. Which is what the paper title is. At present it appears that the Field Analogue part is just an afterthought.

In response to a request by the editor, we have kept the section on the Pozzolane Rosse. We take on board the comments here and have substantially rewritten this section. We have included more data to support our interpretations and have strengthened our case for how the PR structures may be comparable to those in our experiments. We have referred more consistently to the extensive studies on the PR that underlie many of our interpretations presented here and have summarised these where possible.

3. There are many places in the paper where it would benefit enormously from more detail and/ or more extensive discussion (as indicated below in specific points). At present many points, issues etc don't get the discussion they need to and many of the arguments end up appearing cursory and not fully discussed.

We have tried to draw our arguments much more comprehensively and respond to specific points below.

Specific problems

Line 60 Ignimbrite – a widely used term in volcanology but it needs to be defined carefully here. Some of the references given here are for bedforms generated in deposits which would not be defined as ignimbrite by many volcanologists. For example I don't think many Volcanologists would term thinly bedded PDC deposits associated with small tuff cones as ignimbrite. So for this reason it is essential to define exactly what you mean by ignimbrite.

Replaced "ignimbrite" with "PDC deposit" throughout the manuscript unless referring specifically to the Pozzolane Rosse.

Line 71 – The regressive / progressive subdivision of bedforms comes from the sedimentology literature. Allen (1984) should be cited here. Even in the volcanological literature it didn't start with Brand et al 2016. So I think that the literature can be given a bit more acknowledgement here.

This paragraph has been rewritten to provide more background information (lines 61-77)

Line 81 – Quasi steady currents - this is proposed without any justification as far as I can see. Perhaps I am missing something in the odd setup of the journal with methods etc given later, but if there is further information please give it, as at present the reader is just left wondering.

Mass flux variation in these types of experiments has been quantified in Rowley et al 2014. The quasi-steady nature is a result of spontaneous small-scale unsteadiness developing within flows supplied at constant mass flux. The use of quasi steady derives from the use of this term by Branney and Kokelaar (1992, 2002). This has been explained on line 123-124.

Line 84 – don't think its necessary to say 'for the first time' again here. You said that 8 lines back.

Deleted the previous occurrence of the phrase.

Line 99 – Perhaps a few more lines on what the drawbacks might be of not producing 'trains of bedforms', why are these not produced when they are in nature? This could be developed in the discussion a bit more.

Text about the reasons for this has been added to the methods (line 361-364)

Line 108 – stating > 20o is certainly an over generalisation, some are 90o and some look almost 'overturned'

This sentence is now a table, but this has been acknowledged (line 116).

Line 111 – I think there could be a sentence or two of introduction here. It launches straight in.

Introductory sentence added (line 118-119).

Line 114 – '...more rapid deaeration' another term dropped in without any info. The reader is kind of left high and dry...More explanation is needed.

The end of the introduction now has extra methodology information so that the reader can more readily understand the experimental procedure without resorting to the methods (line 90-97).

Line 162 – Field Analogue

The first sentence of this section is a bold unsubstantiated statement and is unacceptable. The field evidence needs to be presented first, then an interpretation arrived at. The reader needs much more descriptive information. The interpretation of these type of deposits is just that an interpretation and we need more than 'matrix supported and fines-depleted' to reach a reasoned conclusion. Its absolutely vital to the understanding. Fines-depleted is a poor term (it also implies an interpretation, how do you know there was fines in the first place?). Why not use 'fines poor' which is a descriptive term.

We have revised the entire section on the Pozzolane Rosse ignimbrite, as explained above. We agree that fines poor is a better term and have replaced 'fines-depleted', except where quoting the field in the graph of Walker (1983).

Line 162 Delete 'and in places partially lithified by zeolites' . It's irrelevant to the story here.

Deleted.

Line 169 Grainsize of 6 samples..... how do we know these are representative of anything? There is no information on the location, facies, stratigraphic position as far as I can see.

Stratigraphic information is now included. The grainsize characteristics are indeed limited due to sampling constraints, but correlate with more extensive grainsize analyses conducted by Giordano and Dobran 1994, as now indicated (line 183-185, fig. 6).

Line 176 -179 So clearly these natural deposits have different properties to the experimental setup. Is this not a serious drawback? This must have some further discussion somewhere. Owing to its importance in terms of the differences between experiments and field.

The higher friction of the field material is countered in the scaling by the correspondingly larger grainsize. Text has been added to explain this (line 189-193).

Line 180 – but you state that it isn't a massive ignimbrite.....How the massive parts relate to the dune bedforms is absolutely critical. At present this information is absent

Massive ignimbrite grades laterally into 'undulated' ignimbrite and has the same deposit characteristics. Text added to explain this (lines 209-213, 275-278)

Line 180 – 185 This paragraph is just far too cursory to address this contentious and complex issue. It absolutely cannot be dealt with in 5.5 lines. What is provided is nowhere nearly enough information.

We have further justified our interpretation of the PR ignimbrite being deposited by a dense PDC on lines 182-185, 209-213, 264-268, 274-278.

Line 187 and Figures 7a and c . These photographs are not clear enough in my opinion, they need to be much clearer. In particular 7c could be anything. The upstream beds are much steeper than 20o. So are your experimental ones. Why not be more specific and give actual angles ?

We have changed figure 7 and used different field photos that better show these bedforms. Angles are shown on fig. 7b and we would prefer to avoid listing numbers in the text where possible.

Line 191 is there a reference for these other bedforms that are found in the Pozzolane Rosse? If so please cite it here.

The shallow stoss-side bedforms referred to are now described directly in the text (line 203-206) and in fig. 7d.

Line 215 In the photographs some parts look like they are 'overturned' in places (Fig 9 d and E). I'm not sure I have seen this in Field analogues. Perhaps you can expand on the similarities and differences more here ? Give more references to those field examples which show these similarities perhaps?

Text explaining the lack of overturned beds in the field analogues has been added (line 200-203).

Line 228 The field analogue certainly doesn't support this 'experimental supposition that dune bedforms can be formed by dense granular currents. It could do if there was adequate careful description and interpretation of what is present in the field, rather than a couple of overly cursory sentences.

We have added more field data and have strengthened our interpretation that the Pozzolane Rosse PDC was a dense, granular current - lines 182-185, 209-213, 264-268, 274-278.

Line 251 – generally apart from where the term regressive is introduced there are no progressive bedforms mentioned. Presumably as they were not generated in the experiments however perhaps there could be more discussion about why you are not seeing/generating progressive bedforms in these experiments and the significance of this.

Sentence about absence of progressive bedforms added (Line 216-217).

Line 264 - I think the suggestion that dune bedforms can be formed by dense flows is demonstrated by this paper. However there are such a wide range of bedforms in reality, in terms of size, wavelength etc that more than just grainsize characteristics are needed. Detailed field relations must be the most important feature to distinguish the two different end members.

More information on field relations has been added in the text (line 209-213, 274-278), as well as been acknowledged in the concluding paragraph (line 313).

References mentioned

Bandi, M. M. et al. Fragility and hysteretic creep in frictional granular jamming. *Phys. Rev. E* **87**, 042205 (2013).

Branney, M. J. & Kokelaar, P. Pyroclastic density currents and the sedimentation of ignimbrites. *Geol. Soc. London. Memoir* **27** (2002).

Branney, M. J. & Kokelaar, P. A reappraisal of ignimbrite emplacement: changes from particulate to non-particulate flow during progressive aggradation of high-grade ignimbrite. *Bull. Volcanol* **54**, 504-520 (1992).

Douillet, G. A. et al. Dune bedforms produced by dilute pyroclastic density currents from the August 2006 eruption of Tungurahua volcano, Ecuador. *Bull. Volcanol.* **75**, 762 (2013).

Douillet, G. et al. Pyroclastic dune bedforms: macroscale structures and lateral variations. Examples from the 2006 pyroclastic currents at Tungurahua (Ecuador). *Sedimentology* (2018).

Giordano, G. & Dobran, F. Computer simulations of the Tuscolano Artemisio's second pyroclastic flow unit (Alban Hills, Latium, Italy). *J. Volcanol. Geoth. Res.* **61**, 69-94 (1994).

Jiang, A. et al. Robotic Granular Jamming: Does the Membrane Matter? *Soft Robot.* **1** (2014).

Silbert, L. E. et al. Analogies between granular jamming and the liquid-glass transition. Phys. Rev. E 65, 051307 (2002).

Walker, G. P. L. Ignimbrite types and ignimbrite problems. J. Volcano. Geotherm. Res. 17, 65-88 (1983).

Walker, G. P. L. Characteristics of dune-bedded pyroclastic surge bedsets. J. Volcanol. Geotherm. Res. 20, 281–296 (1984).

Wohletz, K. H. & Sheridan, M. F. A model of pyroclastic surge. Geol. Soc. Am. Sp. Pap. 180, 177–194 (1979).

Reviewers' comments:

Reviewer #1 (Remarks to the Author):

The authors describe how different bedforms can be generated in dense granular currents, use their results to construct a diagram predicting what bedforms should be generated as a function of current parameters. Finally, they apply their results and diagram to a natural deposit to estimate the flow parameters of the pyroclastic density current that formed the deposit.

The authors addressed nearly of the comments or concerns raised by the reviewers. I should note that the authors attempted to address the few remaining items but they need to go a little bit further. If the items listed below can be address, I recommend the manuscript for publication.

1) Approximately line 57. I recommend including the original references (27-31) as even though they are not about volcanoes, they have definitely informed much of our understanding and interpretation of volcanic deposits and processes.

2) Reviewer 3 raises a good point about comparing experiments and reality (approximately lines 189-203). The new text about grain differences helps, but does not fully explain the relationship between the two. I suggest adding in the range of Savage numbers N_s for the experiments and natural deposits. This will help to show that the experimental and natural systems have similar scaling.

3) Reviewer 3 had questions about "quasi-steady currents" (lines 123-124). The authors have partly addressed this, but it would be useful to add in a single sentence or phrase describing what "quasi-steady" actually means (i.e., "currents with variation in mass flux occurring over a characteristic timescale longer than or comparable to XXXXXXXXXX.")

4) Fig. 5 gran size data – I can tell that the different samples (a-f) have different distributions, but they are presently hard to compare. Could they be combined into a single panel with each line a different color and still labeled a-f?

Best wishes,

Benjamin Andrews, Smithsonian Global Volcanism Program

Reviewer #2 (Remarks to the Author):

Two fundamental things need modifications to my opinion:

1. I still don't really get from the main text where exactly the velocity measurements (and thus Froude numbers etc...) have been taken... Directly at the point of deposition or just before (and thus

at a mobile spot)? at a fixed point after release? Is it for the initial front or for the body of the current?

2. The last paragraph states that backset bedforms are taken as diagnostic for dilute PDCs. I strongly

disagree. It's the organization of deposits as cross-stratification (i.e. with sets of mm to cm scale lamina) in general that is taken as diagnosis for dilute (turbulent) flow-bed boundary conditions... Many cross-laminated deposits show backset beds, but this is not the feature that is calling dilute conditions... (e.g. lines 56 and conclusion from 308). This misleading statement kind of oversell the

outcomes although its not necessary.

A few recurrent phrasings I'd recommend to adapt:

-I find the introduction of figures heavy sometimes (e.g. 111, 195-208),

-I'd avoid the use of "ballotini" and replace by "particles". (I don't know what ballotini are)

-In several occurrences, you list first the steep backset data, then shallow, then flat. This is reversed in term of chronology of deposition, and deceleration of the flow. I'd follow the chronology (L136, L164, etc..)

-Sample description (granulometry etc...): I'd rephrase, see text.

-Your field examples are at Pozzolane Rosso, this is clear from one sentence. No need to repeat "Pozzolane Rosso" that often (17 occurrences...)

Generally, I'd suggest to shift part of the main text in the captions (see line by line)

-Lines 211, 249-252, 272-285, 286-289 all deal with the question of the massive/backset bedded question and interpretation of dense granular flows. As they come, it feels repetitive, and they should be arranged together in a strong chapter (at the very beginning of the discussion). Little things I did not like but figure its the editor choice:

-I guess the editors and other reviewers wanted more field details... I still disagree with this, but that's only my opinion, so fine.

-Does the Method section really have to go at the end? (and figure 10 be last instead of first?)

Figures:

Figure 2: Please find a way to get all images centered on the aeration limit at 1 m. The way it is presented here is misleading and structures cannot be compared. I'd rather crop-out part of a proximal/distal end than get confused on the deposition onset. vs deaeration limit..

Figure 5: Can you put all your cumulative curves on a single graph? I don't see much difference between samples, neither a fundamental use of these curves so I would minimize the space it takes...

Figure 6: I think there's little reason to have figure 6 detached from figure 5, could you do it figure 5c instead?

Captions: see line by line

Line by line:

33: is "by" correct or should it be through an upstream propagating? (not native question)

41: is there any reference for this first sentence?

42: Deposit from deep see turbidity currents form... (it refers to the sediment, not the flow here)

41-44: "PDCs" should be introduced at the first occurrence (41 rather than secon in 44

46: seafloor infrastructure(s?) => isn't it plural (not native)?

50: internal dynamics? (instead of interior dynamics)

56: I'd use cross-stratified/cross-laminated bedforms (Very coarse and massive bedforms can result

from a dense pyroclastic flow...)

61: maybe you can state somewhere here what are backset structures (i.e. beds deposited on the upstream side of an obstacle)? I doubt it is clear to every reader what backsets are, and it is so central that you might want to be sure that the definition is clear and vulgarized...

63 which are "generally thought to be" formed under supercritical [...]

85, 118, 190: I do not know the word "ballotini", not sure it is commonly known... Could you put "fine particles" instead and simply state in the section "Experimental material" at line 339 "the experiments were performed using PARTICLES OF spherical soda lime ballotini..."

87: As deposition aggrades... I'd say either "as deposit aggrades" or "As deposition occurs"

88: same, either "recorded using a high speed camera" or "recorded on high speed videos"

94: rapid deaeration "hypothesized" to occur instead of "we know"?

95: coarser instead of courser (not native)?

106 either "very shallow backsets" or "very shallowly backsetting"

109: where steep and shallow backset bedforms are present... unclear to me, you mean "where there

are both steep sided and shallow sided backsets beds occurring in a single bedform"? or that "for both the steep backset type and the shallow backset type", a group of 3-4 stoss-side lamina

merge

into a single laminae on the lee side?

110-111: not sure a new paragraph is needed here (and the 109-110 paragraph is a single sentence

right now)

110: I suggest "dipping at varying angle and converging/merging into a single corresponding leese side laminae.

111: I would just add (Tab 1) at the end of line 110 in brackets and remove sentence in 111: "Characteristics of steep and shallow backset bedforms are given in Table 1."

112: Similarly, I think this sentence is not needed, since you introduced the figure in line 106: "The

experimental deposits are presented in figure 2"

118: "ballotini", replace by particles

135-138: I suggest reversing the order of listing to keep the chronology of deposition: first explain the planar beds, then the shallow backsets, and end up with the steep ones.

140: "As a result of thickening", is this within a single flow or between experimental runs of variable thicknesses?

151: N_f is a ratio instead of the ratio?

162: by definition of $Fr = U/(gH)^{0.5}$, isn't it normal that U is correlated to Fr ?

164-166: again, I recommend to reverse the order of listing to begin with the onset of deposition and follow the deceleration chronology.

167: no need for a new paragraph, this concludes on the data just listed... (and I find 1-sentence paragraphs are not very elegant)

176-184: I suggest rephrasing to shorten and reorganize the structure of appearance (first where samples come from, then how they look like, then what's the particle size distribution, and finally what it means). Here a suggestion:

"Six samples were taken from three localities (within 18-24 km of the vent) and two facies (massive, and 'undulated' bedding as described in Giordano & Doronzo63, Fig. 5). Grains are dominantly poorly vesicular scoria with compositions plotting in the tephrite/basanite field62. The grain size analysis shows that distribution of all samples is are dominated by lapilli-sized grains and

poor in the < 63 μm fraction (Fig. 5), and are consistent with samples from other studies (Fig. 6). All plot in the 'fines-depleted flow' field of Walker25, close to Pozzolane Rosse samples taken 18-24 km from the vent, analysed by Giordano & Dobran58 (Fig. 6). Hence, no differences in differences in grain size distribution at localities where the ignimbrite is massive vs where it contains bedforms was observed and the Pozzolane Rosse ignimbrite is considered a valid example of for an analogue dense granular current."

187: suggest to delete "on the six samples taken from the Pozzolane Rosse": the entire chapter is about Pozzolane Rosse, no need to repeat where they come (also at former line 176)

190: replace ballotini with "particles"

192: I am lost, grain size does not appear in the scaled equations, so what do you mean with this scaling? They have similar N_f ?

195: mixing flow and deposits here. the flows are interpreted to have left the plain and run up, but the ignimbrite IS there, not moving. Please rephrase.

195-198: I suggest to keep this minimal in the text and keep details in the caption. The natural examples (Fig 7) share similarities with our experiments (e.g. stoss sides for fig 7a vs 2a-c, lee in 7c vs 2d).

205: keep the text in brackets for the caption and just put (Fig 7d): "the example in Figure 7d is thicker by ~ 15 cm over the stoss and crest compared to the lee"

207: Again a sentence just to call a figure, and reminding that you talk about Pozzolane Rosso... It is

to my opinion non-aesthetic and heavy for the reader. Maybe something like "Measured stoss angles

span the same range as experimental deposits, yet the lee beds are much steeper (Fig 7b)"

209: grades laterally? suggest "evolves" instead of grade, which may be connoted as a grain size

evolution. "laterally"? could you say "downflow/ towards paleovalleys/ towards depocenters?"

210: suggest to add (Fig5-6) at the end of the sentence

211: Careful, this belongs to the interpretation (and is developed at line 279 and onward already). It

is out of place and creates a repetition with later statements. Also, I'm not sure this "it is reasonable

to infer..." is very scientific... Maybe something like "It is interpreted here that the same event deposited both facies without significant changes in dynamics, such as particle concentration or turbulence level." Again, this would belong to the interpretation (at least move it just below the "Discussion" title?)

214: I would begin the discussion with a paragraph about the fact that your granular flows produce bedforms, thus backset bedforms are not only related to dilute PDCs, that the two facies of your field examples have similar grain sizes, and that you don't interpret a change in dynamics between both facies. This is probably the priority of your message, and first thing to understand. You could do so by using lines 249-252, this sentence line 211, and the two paragraphs 272-285 (for example just shifting lines 249-252 here, removing the parts on the Froude jump). Then you directly follow that your experiments supports that the same event and dynamics can create the massive and undulated beds in the field examples from the red Pozzolane.

215-217: The two first sentences are descriptive and belong to the data (move them just before the

"Discussion" title?). The third sentence belongs to the part developed on Froude numbers at lines 223-230 (Maybe simply write after 230 "The absence of downstream propagating structures (progressive) is interpreted as the quasi absence of subcritical conditions in the experiments"?)

222-223: No need to change paragraph, it's all about Froude numbers?

223: add (Fig 4) at end of sentence?

225: I suggest to rephrase as "An apparent Froude jump occurs during deposition of the steep backset beds." If the planar and shallow backset beds are deposited at $Fr > 1$, they cannot experience

a Froude jump, so that the jump is only valid for the steep backsets...

234: [...] to mean the upstream propagation of the "FREEZING / DEPOSITION FRONT" of the granular material

240: [...] steepening "OF STOSS FACIES/BEDS" well beyond the repose angle

250 [...] Froude jumps within "DILUTE" PDCs into question [...]. Also, this is a one sentence paragraph, could you put it somewhere else?

266 [...] close to the fluid-escape "AND GRANULAR-BASED" end member "S" of Branney & Kokelaar (2002)

278-279: keep in a single paragraph, beginning sentence in 279 with "INSTEAD, THE CHANGE IN DACIES COULD BE DUE TO" the onset of [...]

286-289: it becomes very repetitive to me (or I don't see the subtleties?) suppress?

289-291: maybe better after line 296 with slight rephrasing? I suggest: [...] until it collapses upstream by avalanching (Fig 9d-e). "Our work "FURTHER" provides evidences that this process can create bedforms with "dense granular" PDC deposits. These mechanisms support the stoss-side

blocking process suggested by Douillet based on field deposits. "

300: Froude number(S) (remove S?)

301: the backset bedforms are technically recording "the transition from supercritical to subcritical (transcritical) flow"?

304-305: maybe not necessary to mention, the reader can judge by himself (could be seen as arrogant by some readers?).

308: I disagree. It's the presence of cross-stratification/cross-lamination that is used as diagnostic evidence for dilute turbulent currents, not backsets.

570: I suggest "Sketches of backset bedforms from PDC deposits". Also, please remove all "idealised"

613: flow direction is clear from the arrow (no need to write?)

614 I suggest: "b. Stoss vs lee beds angle for the Pozzolane and experimental bedforms." Could all

coordinates go in the supplementary data? They are not fundamental in your study and there's already a location map. (And they need a reference system (WGS84?), and could be easier to understand in degrees minutes seconds...)

Response to Review: Smith et al “A bedform phase diagram for dense granular currents”

We would like to begin by thanking the reviewers and editors for the careful consideration of our paper. Below we outline, in bold, how we have addressed the reviewers’ comments (provided) on a point-by-point basis. Our line numbers refer to the text with ‘no’ or ‘simple’ markup.

Reviewer 1

The authors describe how different bedforms can be generated in dense granular currents, use their results to construct a diagram predicting what bedforms should be generated as a function of current parameters. Finally, they apply their results and diagram to a natural deposit to estimate the flow parameters of the pyroclastic density current that formed the deposit. The authors addressed nearly of the comments or concerns raised by the reviewers. I should note that the authors attempted to address the few remaining items but they need to go a little bit further. If the items listed below can be address, I recommend the manuscript for publication.

We thank the reviewer for these comments and recognition of the work we have undertaken in addressing the original review.

1) Approximately line 57. I recommend including the original references (27-31) as even though they are not about volcanoes, they have definitely informed much of our understanding and interpretation of volcanic deposits and processes.

These references are now numbered 15-19. (Line 55)

2) Reviewer 3 raises a good point about comparing experiments and reality (approximately lines 189-203). The new text about grain differences helps, but does not fully explain the relationship between the two. I suggest adding in the range of Savage numbers N_s for the experiments and natural deposits. This will help to show that the experimental and natural systems have similar scaling.

We have made this change and the section reads: “ N_s in these experiments range from 0.00003-0.03, and N_B from 15-269. In natural PDCs, N_s range from 10^{-8} - 10^{-9} ¹³, which like our experiments is in the frictional regime⁵⁶ despite the difference of several orders of magnitude. Our N_B values overlap with those of natural PDCs (10^0 - 10^2)¹³.” (Line 161-164)

3) Reviewer 3 had questions about “quasi-steady currents” (lines 123-124). The authors have partly addressed this, but it would be useful to add in a single sentence or phrase describing what “quasi-steady” actually means (i.e., “currents with variation in mass flux occurring over a characteristic timescale longer than or comparable to XXXXXXXXXXXX.”)

We have made the following change – introducing text that states: “Small spontaneously-generated variations in the current mass flux result in minor unsteadiness in the flow over timescales in the order of 0.05 s and flow thickness variations in the order of +/- 10%, hence their quasi or nearly-steady nature⁴⁴.” (Lines 121-124)

4) Fig. 5 gran size data – I can tell that the different samples (a-f) have different distributions, but they are presently hard to compare. Could they be combined into a single panel with each line a different color and still labeled a-f?

Yes, this change has been made.

Reviewer 2

1. I still don't really get from the main text where exactly the velocity measurements (and thus Froude numbers etc...) have been taken... Directly at the point of deposition or just before (and thus at a mobile spot)? at a fixed point after release? Is it for the initial front or for the body of the current?

We have added the sentence: "Reported velocities are calculated by tracking coloured sediment packages in the body of the current immediately prior to their deposition." (Lines 354-356)

2. The last paragraph states that backset bedforms are taken as diagnostic for dilute PDCs. I strongly disagree. It's the organization of deposits as cross-stratification (i.e. with sets of mm to cm scale lamina) in general that is taken as diagnosis for dilute (turbulent) flow-bed boundary conditions... Many cross-laminated deposits show backset beds, but this is not the feature that is calling dilute conditions... (e.g. lines 56 and conclusion from 308). This misleading statement kind of oversell the outcomes although its not necessary.

We have changed these occurrences to refer to cross-stratification and bedforms in general rather than just backsets. (Lines 56 + 302)

A few recurrent phrasings I'd recommend to adapt:

-I find the introduction of figures heavy sometimes (e.g. 111, 195-208),

The text referenced has been changed so that figures and tables are called more naturally. (Lines 112 + 201-204)

-I'd avoid the use of "ballotini" and replace by "particles". (I don't know what ballotini are)

Replaced as suggested. (Lines 86, 117, 196)

-In several occurrences, you list first the steep backset data, then shallow, then flat. This is reversed in term of chronology of deposition, and deceleration of the flow. I'd follow the chronology (L136, L164, etc..)

Text rewritten as suggested. (Lines 125-130, 171-172)

-Sample description (granulometry etc...): I'd rephrase, see text.

Rephrased as suggested. (Lines 183-190)

-Your field examples are at Pozzolane Rosso, this is clear from one sentence. No need to repeat "Pozzolane Rosso" that often (17 occurrences...)
Generally, I'd suggest to shift part of the main text in the captions (see line by line)

Repeat occurrences have been shortened to "PR" throughout. PR is retained because this is the name of the ignimbrite as defined, and not just a reference to the location of the field study.

-Lines 211, 249-252, 272-285, 286-289 all deal with the question of the massive/backset bedded question and interpretation of dense granular flows. As they come, it feels repetitive, and they should be arranged together in a strong chapter (at the very beginning of the discussion).

The discussion has been rewritten to be more succinct and rearranged to make the argument clearer.

Little things I did not like but figure its the editor choice:

-I guess the editors and other reviewers wanted more field details... I still disagree with this, but that's only my opinion, so fine.

-Does the Method section really have to go at the end? (and figure 10 be last instead of first?)

Nature Comms format has Methods at the end.

Figures:

Figure 2: Please find a way to get all images centered on the aeration limit at 1 m. The way it is presented here is misleading and structures cannot be compared. I'd rather crop-out part of a proximal/distal end than get confused on the deposition onset. vs deaeration limit..

All images are now centred on the 1 m mark.

Figure 5: Can you put all your cumulative curves on a single graph? I don't see much difference between samples, neither a fundamental use of these curves so I would minimize the space it takes...

All curves are now on the same graph.

Figure 6: I think there's little reason to have figure 6 detached from figure 5, could you do it figure 5c instead?

Figure 6 is now Figure 5c.

Line by line:

33) : is "by" correct or should it be through an upstream propagating? (not native question)

Sedimentation 'by' a bore is a common phrase in the literature so we have left as is.

41: is there any reference for this first sentence?

This is an introductory sentence. Specific examples supporting it are given, with references, in the following sentences.

42: Deposit from deep see turbidity currents form... (it refers to the sediment, not the flow here)

Sentence changed to: "Deep-sea turbidity currents deposit the largest sediment accumulations on Earth..." (Line 42)

41-44: "PDCs" should be introduced at the first occurrence (41 rather than second)

Line 41 refers to particulate density currents in general, such as turbidity currents, rather than PDCs specifically.

46: seafloor infrastructure(s?) => isn't it plural (not native)?

We think “infrastructure” is appropriate as we are referring to a singular place (the seafloor).

50: internal dynamics? (instead of interior dynamics)
Changed to “internal”. (Line 50)

56: I'd use cross-stratified/cross-laminated bedforms (Very coarse and massive bedforms can result from a dense pyroclastic flow...)
Sentence changed to “Various types of cross-stratified bedforms occur...”. (Line 56)

61: maybe you can state somewhere here what are backset structures (i.e. beds deposited on the upstream side of an obstacle)? I doubt it is clear to every reader what backsets are, and it is so central that you might want to be sure that the definition is clear and vulgarized...
Added phrase “(i.e. upstream-dipping beds)”. (Line 62)

63 which are "generally thought to be" formed under supercritical [...]
Added “generally thought to be”. (Line 64)

85, 118, 190: I do not know the word "ballotini", not sure it is commonly known... Could you put "fine particles" instead and simply state in the section "Experimental material" at line 339 "the experiments were performed using PARTICLES OF spherical soda lime ballotini..."
Replaced “ballotini” with “particles where suggested (Lines 86, 117, 196) and added “particles of” (Line 340).

87: As deposition aggrades... I'd say either "as deposit aggrades" or "As deposition occurs"
Changed sentence to “As the deposit aggrades”. (Line 88)

88: same, either "recorded using a high speed camera" or "recorded on high speed videos"
Changed sentence to “... using a high-speed camera”. (Line 89)

94: rapid deaeration "hypothesized" to occur instead of "we know"?
Changed sentence to “...rapid deaeration hypothesised to occur”. (Line 94)

95: coarser instead of courser (not native)?
Changed to “coarser”. (Line 95)

106 either "very shallow backsets" or "very shallowly backsetting"
Changed to “...very shallow backset...”. (Line 106)

109: where steep and shallow backset bedforms are present... unclear to me, you mean "where there are both steep sided and shallow sided backsets beds occurring in a single bedform"? or that "for both the steep backset type and the shallow backset type", a group of 3-4 stoss-side lamina merge into a single laminae on the lee side?

Sentence changed to: "Both steep and shallow backset bedforms comprise a bedset of...". (Line 110)

110-111: not sure a new paragraph is needed here (and the 109-110 paragraph is a single sentence right now)

The new paragraph referred to no longer exists.

110: I suggest "dipping at varying angle and converging/merging into a single corresponding leeside laminae.

Changed to "...converging into a single corresponding lee-side lamina." (Line 111)

111: I would just add (Tab 1) at the end of line 110 in brackets and remove sentence in 111: "Characteristics of steep and shallow backset bedforms are given in Table 1."

"(Table 1)" added as suggested (Line 112), and sentences deleted as suggested.

112: Similarly, I think this sentence is not needed, since you introduced the figure in line 106: "The experimental deposits are presented in figure 2"

This sentence has been deleted, and the following sentence moved to lines 108-109.

118: "ballotini", replace by particles

Replaced as suggested. (Line 117)

135-138: I suggest reversing the order of listing to keep the chronology of deposition: first explain the planar beds, then the shallow backsets, and end up with the steep ones.

The listing of bedforms has been reversed as suggested. (Lines 125-130)

140: "As a result of thickening", is this within a single flow or between experimental runs of variable thicknesses?

Within a single flow. Sentence rewritten to: "As a result of thickening within a steady current bedform-induced deposits of different character can be formed..." (Lines 141-142)

151: N_f is a ratio instead of the ratio?

We prefer the phrasing "is the ratio", and we define it in the next line.

162: by definition of $Fr = U/(gH)^{0.5}$, isn't it normal that U is correlated to Fr ?

True, added "As anticipated..." before this sentence. (Line 168). It's important that this correlation is stated and that our flows are behaving as expected. However, this correlation isn't as strong at higher velocities, as also stated.

164-166: again, I recommend to reverse the order of listing to begin with the onset of deposition and follow the deceleration chronology.

The listing of bedforms has been reversed as suggested. (Lines 171-172)

167: no need for a new paragraph, this concludes on the data just listed... (and I find 1-sentence paragraphs are not very elegant)

Paragraph break deleted.

176-184: I suggest rephrasing to shorten and reorganize the structure of appearance (first where samples come from, then how they look like, then what's the particle size distribution, and finally what it means). Here a suggestion: "Six samples were taken from three localities (within 18-24 km of the vent) and two facies (massive, and 'undulated' bedding as described in Giordano & Doronzo63, Fig. 5). Grains are dominantly poorly vesicular scoria with compositions plotting in the tephrite/basanite field62. The grain size analysis shows that distribution of all samples is are dominated by lapilli-sized grains and poor in the < 63 μm fraction (Fig. 5), and are consistent with samples from other studies (Fig. 6). All plot in the 'fines-depleted flow' field of Walker25, close to Pozzolane Rosse samples taken 18-24 km from the vent, analysed by Giordano & Dobran58 (Fig. 6). Hence, no differences in differences in grain size distribution at localities where the ignimbrite is massive vs where it contains bedforms was observed and the Pozzolane Rosse ignimbrite is considered a valid example of for an analogue dense granular current."

This paragraph has been rewritten taking into account the above comments. (Lines 183-190)

187: suggest to delete "on the six samples taken from the Pozzolane Rosse": the entire chapter is about Pozzolane Rosse, no need to repeat where they come (also at former line 176)

Sentence in question now reads: "Six samples were taken for this study from three localities (within 18-24 km of the vent; Fig. 5a) and two facies..." (Lines 183-184). Repeated instances of "Pozzolane Rosse" throughout the manuscript have been replaces by "PR". PR is retained because this is the name of the ignimbrite as defined, and not just a reference to the location of the field study.

190: replace ballotini with "particles"

Replaced as suggested. (Line 194)

192: I am lost, grain size does not appear in the scaled equations, so what do you mean with this scaling? They have similar N_f ?

Grain size does appear in the referred to Eq. 2 as δ (particle/grain diameter). (Line 157)

195: mixing flow and deposits here. the flows are interpreted to have left the plain and run up, but the ignimbrite IS there, not moving. Please rephrase.

Rephrased as: "...where the depositing current left the radial plain and ran up...". (Line 199)

195-198: I suggest to keep this minimal in the text and keep details in the caption. The natural examples (Fig 7) share similarities with our experiments (e.g. stoss sides for fig for the caption and just put (Fig 7d): "the example in Figure 7d is thicker by ~15 7a vs 2a-c, lee in 7c vs 2d).

Detailed text has been deleted. Sentence rewritten as: "The bedforms in the PR share similarities with our experimental deposits (c.f. Fig. 6a and Fig. 2a-c, Fig. 6c and Fig. 2d)" (Line 203-204)

205: keep the text in brackets cm over the stoss and crest compared to the lee"
"is thicker by ~15 cm over the stoss and crest compared to the lee" has been moved to the figure caption. (Line 620)

207: Again a sentence just to call a figure, and reminding that you talk about Pozzolane Rosso... It is to my opinion non-aesthetic and heavy for the reader. Maybe something like "Measured stoss angles span the same range as experimental deposits, yet the lee beds are much steeper (Fig 7b)"

Sentence rewritten as "...measured stoss angles for both natural and experimental bedforms span the same range (Fig. 6b)" and moved to lines 204-205.

209: grades laterally? suggest "evolves" instead of grade, which may be connoted as a grain size evolution. "laterally"? could you say "downflow/ towards paleovalleys/ towards depocenters?"

Evolves has connotations of time, which given that this is lateral (e.g. along-deposit), this would be incorrect. We have changed 'grades' to 'transitions' (line 200). We have retained 'laterally' as the transition is not as consistent as 'downflow/towards palaeovalleys/towards depocenters'.

210: suggest to add (Fig5-6) at the end of the sentence

We think (Fig. 5) is better called in the next sentence. (Line 202)

211: Careful, this belongs to the interpretation (and is developed at line 279 and onward already). It is out of place and creates a repetition with later statements. Also, I'm not sure this "it is reasonable to infer..." is very scientific... Maybe something like "It is interpreted here that the same event deposited both facies without significant changes in dynamics, such as particle concentration or turbulence level." Again, this would belong to the interpretation (at least move it just below the "Discussion" title?)

The sentence containing interpretation has been deleted. The preceding two sentences have been moved to lines 200-202.

214:I would begin the discussion with a paragraph about the fact that your granular flows produce bedforms, thus backset bedforms are not only related to dilute PDCs, that the two facies of your field examples have similar grain sizes, and that you don't interpret a change in dynamics between both facies. This is probably the priority of your message, and first thing to understand. You could do so by using lines 249-252, this sentence line 211, and the two paragraphs 272-285 (for example just shifting lines 249-252 here, removing the parts on the Froude jump). Then you directly follow that your experiments supports that the same event and dynamics can create the massive and undulated beds in the field examples from the red Pozzolane.

The discussion has been rewritten to more succinct and rearranged to make the argument clearer in response to other comments by the reviewers. The focus of the paper is on the experimental data, so we would prefer the

discussion to stay focussed on this data, rather than discuss the field example up front as suggested here.

215-217: The two first sentences are descriptive and belong to the data (move them just before the "Discussion" title?). The third sentence belongs to the part developed on Froude numbers at lines 223-230 (Maybe simply write after 230 "The absence of downstream propagating structures (progressive) is interpreted as the quasi absence of subcritical conditions in the experiments"?)

We prefer the first sentence where it is as a general introduction. As progressive bedforms are not created in these experiments, we have removed them entirely from the discussion to make it more succinct and focussed. A statement on why they are not formed is given when we state we do not create them (lines 112-113).

222-223: No need to change paragraph, it's all about Froude numbers?
Paragraph break deleted.

223: add (Fig 4) at end of sentence?
Added as suggested. (Line 220)

225: I suggest to rephrase as "An apparent Froude jump occurs during deposition of the steep backset beds." If the planar and shallow backset beds are deposited at $Fr > 1$, they cannot experience a Froude jump, so that the jump is only valid for the steep backsets...

Rephrased as: "...an apparent Froude jump within the flow forms in the current during deposition of the steep backset bedforms..." (Line 222-223)

234: [...] to mean the upstream propagation of the "FREEZING / DEPOSITION FRONT" of the granular material
Added "...propagation of the depositional front of the granular material". (Line 232-233)

240: [...] steepening "OF STOSS FACES/BEDS" well beyond the repose angle
Added "steepening of stoss faces well beyond..." (Line 238)

250 [...] Froude jumps within "DILUTE" PDCs into question [...]. Also, this is a one sentence paragraph, could you put it somewhere else?
Added "dilute" as suggested (line 248) and added this sentence to the end of the preceding paragraph.

266 [...] close to the fluid-escape "AND GRANULAR-BASED" end member "S" of Branney & Kokelaar (2002)
Added as suggested. (Line 264)

278-279: keep in a single paragraph, beginning sentence in 279 with "INSTEAD, THE CHANGE IN DACIES COULD BE DUE TO" the onset of [...]
Added link between paragraphs as suggested. (Line 277)

286-289: it becomes very repetitive to me (or I don't see the subtleties?) suppress?

Agreed, the first sentence of the paragraph has been deleted and the second moved to and rewritten in lines 288-290 as suggested below.

289-291: maybe better after line 296 with slight rephrasing? I suggest: [...] until it collapses upstream by avalanching (Fig 9d-e). "Our work "FURTHER" provides evidences that this process can create bedforms with "dense granular" PDC deposits. These mechanisms support the stoss-side blocking process suggested by Douillet based on field deposits. "

Sentence moved to lines 288-290 and rewritten as: "Our work provides direct evidence that bedforms can be created by dense granular PDCs, and supports the stoss-side blocking process first suggested by Douillet^{22,70} based on field deposits."

300: Froude number(S) (remove S?)
Removed as suggested. (Line 294)

301: the backset bedforms are technically recording "the transition from supercritical to subcritical(transcritical) flow"?
Added "the transition from supercritical to subcritical". (Line 295)

304-305: maybe not necessary to mention, the reader can judge by himself (could be seen as arrogant by some readers?).
We think that it is important that the application of this work is clearly articulated, so we have left as is.

308: I disagree. It's the presence of cross-stratification/cross-lamination that is used as diagnostic evidence for dilute turbulent currents, not backsets.
Added "(e.g. cross-stratification and backsets)". (Line 303)

570:I suggest"Sketches of backset bedforms from PDC deposits". Also, please remove all "idealised"
Replaced "Examples" with "Sketches" and removed all "idealised". (Lines 578-581)

613: flow direction is clear from the arrow (no need to write?)
Removed as suggested. (Line 615)

614 I suggest: "b. Stoss vs lee beds angle for the Pozzolane and experimental bedforms." Could all coordinates go in the supplementary data? They are not fundamental in your study and there's already a location map. (And they need a reference system (WGS84?), and could be easier to understand in degrees minutes seconds...)

Added: "b stoss and lee angles for PR and experimental backset bedforms", and "using the WGS84 Datum" (lines 615-617). We would prefer to keep the coordinates in the captions (they don't take up much space and are easier for readers to find).